



# Structure and drivers of ocean mixing north of Svalbard in summer and fall 2018

Zoe Koenig[1,2], Eivind H. Kolås[1], and Ilker Fer[1]

[1]Geophysical Institute, University of Bergen and Bjerknes Center for Climate Research, Bergen, Norway
[2]Norwegian Polar Institute, Tromsø, Norway

**Correspondence:** Zoe Koenig (zoe.koenig@uib.no)

**Abstract.** Ocean mixing in the Arctic Ocean cools and freshens the Atlantic and Pacific-origin waters by mixing them with surrounding waters, which has major implications on global scale as the Arctic Ocean is a main sink for heat and salt. We investigate the drivers of ocean mixing north of Svalbard, in the Atlantic sector of the Arctic, based on observations collected during two research cruises in summer and fall 2018. In the mixed layer, there is a nonlinear relation between the layer-integrated dis-

sipation and wind energy input; convection was active at a few stations and was responsible for enhanced turbulence compare to what was expected from the wind work alone. Summer melting of sea ice reduces the temperature, salinity and depth of the mixed layer, and increases salt and buoyancy fluxes at the base of the mixed layer. Deeper in the water column and near the seabed, tidal work is a main source of turbulence: diapycnal diffusivity in the bottom 250 m of the water column is enhanced during strong tidal currents, reaching on average $10^{-3}$ m$^2$ s$^{-1}$. The average profile of diffusivity decays with distance from

seabed with an e-folding scale of 22 m compared to 18 m in conditions with weaker tidal currents. A nonlinear relation is inferred between the depth-integrated dissipation in the bottom 250 m of the water column and the tidally-driven bottom drag, and is used to estimate the bottom dissipation along the continental slope of the Eurasian Basin. Computation of the inverse Froude number suggests that nonlinear internal waves forced by the diurnal tidal activity ($K_1$ constituent) can develop north of Svalbard and in the Laptev and Kara Seas, with the potential to mix the entire water column vertically. Estimates of vertical

turbulent heat flux from the Atlantic Water layer up to the mixed layer reaches 30 W m$^{-2}$ in the core of the boundary current, and is on average 8 W m$^{-2}$, accounting for $\sim 1\%$ of the total heat loss of the Atlantic layer in the region.

## 1 Introduction

The Arctic Ocean is a sink for salt and heat. Relatively warm and salty Atlantic waters enter the Arctic Ocean via Fram Strait

and the Barents Sea through-flow, and colder and fresher Arctic waters exit flowing east of Greenland through the East Greenland Current. Annual average water mass transformation in the Arctic is about $-0.62 \pm 0.23$ in salinity and $-3.74 \pm 0.76°$C in temperature (Tsubouchi et al., 2018). With the rapid and large sea ice decline, the Arctic Ocean is particularly vulnerable to





climate change. In the near future we may enter a new regime, in which the interior Arctic Ocean is entirely ice free in summer and sea ice is thinner and more mobile in winter, which will have vast implications for the Arctic ocean circulation, the marine

ecosystems it supports, and the larger-scale climate (Timmermans and Marshall, 2020). The heat reservoir that contains the Atlantic and Pacific origin waters has the potential to melt the entire sea ice if reaching the surface (Maykut and Untersteiner, 1971). The estimated mean Arctic Ocean surface heat flux necessary to keep the sea ice thickness at equilibrium is $2 \, \mathrm{W \, m^{-2}}$ (Maykut and McPhee, 1995), yet observations indicate mean surface heat fluxes of $3.5 \, \mathrm{W \, m^{-2}}$ (Krishfield and Perovich, 2005). To assess the evolution of the sea ice, the oceanic heat in the Arctic must be monitored and understood.

Atlantic Water is a main component of the Arctic Ocean heat budget, with particular influence in the Atlantic sector. An important player in the transformation of the Atlantic Water is vertical mixing. In the Arctic Ocean, vertical mixing is dominated by turbulence generated by processes over topography and along margins (Padman and Dillon, 1991; Lenn et al., 2011; Rippeth et al., 2015; Fer et al., 2014), while the central Arctic is relatively quiescent (Fer, 2009; Lincoln et al., 2016). Microstructure measurements indicate turbulent kinetic energy dissipation in the halocline of the deep basins to be around $10^{-10}$ to $10^{-9}$

$\mathrm{W \, kg^{-1}}$ (Fer, 2009; Lincoln et al., 2016; Rippeth et al., 2015). The dissipation rates are estimated to be 2 orders of magnitude larger on the ocean margins than over the abyssal plain, for example about $3 - 20 \times 10^{-8} \, \mathrm{W \, kg^{-1}}$ in the region just north of Svalbard (Rippeth et al., 2015).

Vertical mixing in the Arctic Ocean is driven by various energy sources, such as winds, tides and mesoscale activity. Zhao et al. (2014) described a prevalent Arctic eddy field, typically generated by instability of surface fronts (in the eastern Canada

basin) or instability of boundary currents (in the southwestern Canada Basin or in the vicinity of ridge features and in shelf regions in the Eurasian Basin). Topographic waves generated over bathymetric slopes and rough topography, forced by the tides, are the main source of energy for increased tidal dissipation rates observed over topography (Padman et al., 1992; Rippeth et al., 2017; Fer et al., 2020b). Wind-driven momentum input to the Arctic ocean is largely dampened by sea ice cover (Rainville and Winsor, 2008). In winter months with complete ice cover, inertial wave energy and shear are generally weaker

than in summer when we observe an increased atmosphere to ocean momentum transfer in open water regions (Dosser and Rainville, 2016).

North of Svalbard is a location with enhanced mixing in the Arctic. It is also a key region for the Arctic Ocean heat and salt budget, as it is the gateway for the Fram Strait inflow of Atlantic Water. The circulation of Atlantic Water here is complex, with several recirculations in Fram Strait and three main inflow branches including the Yermak Branch (YB, Cokelet et al. (2008)),

the Yermak Pass Branch (YPB, Koenig et al. (2017); Crews et al. (2019); Menze et al. (2019)) and the Svalbard Branch (SB, Cokelet et al. (2008)), all originating from the West Spitsbergen Current (WSC) (Figure 1a). As the Atlantic Water flows eastward, it deepens, gets colder and fresher due to mixing with the surrounding waters.

Cooling and freshening of the Atlantic Water north of Svalbard result from different processes. Along the slope north of Svalbard, eddies are shed from the Atlantic Water Boundary Current (Våge et al., 2016; Crews et al., 2018), transporting

0.16 Sv of Atlantic Water and 1.0 TW away from the boundary current. At depth, the ocean is affected by the tidal work: Rippeth et al. (2015) showed that the Yermak Plateau is a hot spot for tidal mixing and Fer et al. (2014) suggested that in the region almost the entire dissipation can be attributed to the dissipation of baroclinic tidal energy. In the Nansen Basin north of

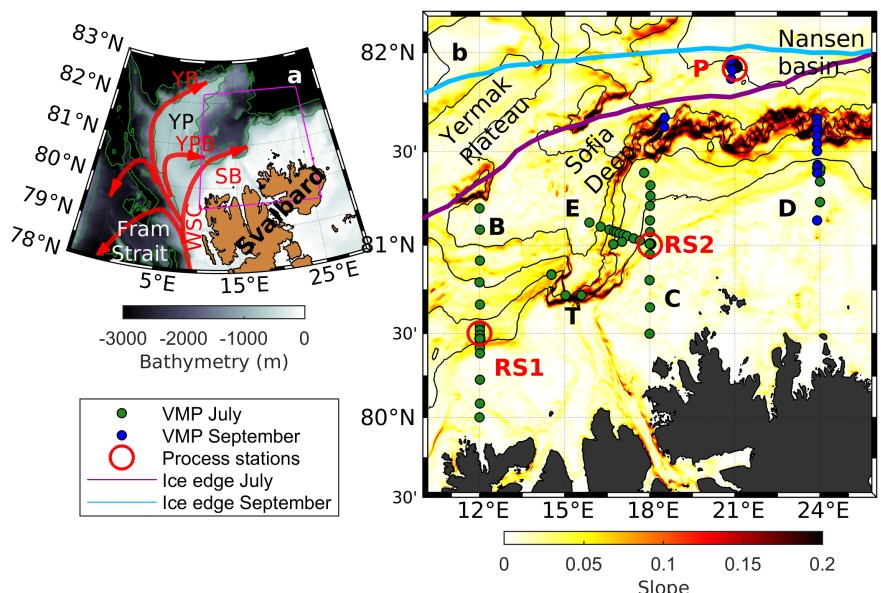

**Figure 1.** a) Circulation pattern of Atlantic Water around Svalbard, with the West Spitsbergen Current (WSC), the Svalbard Branch (SB), the Yermak Branch (YB) and the Yermak Pass Branch (YPB). Bathymetry is from the International Bathymetric Chart of the Arctic Ocean, IBCAO-v3 (Jakobsson et al., 2012). b) Close-up on the magenta box in panel a. Station locations (June, green dots; September, blue dots), sections (B, C, D and E), and process stations (red circles marked RS1, RS2 and P) are shown. The slope steepness calculated from IBCAO-v3 is color-coded at the background. Isobaths are drawn every 1000 m in (a) and 500 m in (b). Purple/light blue lines are the averaged sea ice edge defined as the 15% ice concentration over the summer/fall cruise respectively.

Svalbard, turbulence in the upper layer influences the sea ice cover. Peterson et al. (2017) found an average winter ocean-to-ice heat flux of around 1.4 W m$^{-2}$ , with episodic local upwelling events and proximity to Atlantic Water pathways increasing the

heat fluxes by one order of magnitude. Meyer et al. (2017) presented 6 months of turbulence data collected from January to June 2015 during the N-ICE2015 campaign. The combination of storms and shallow Atlantic Water leads to the highest heat flux rates observed: ice-ocean interface heat fluxes averaged 100 W m$^{-2}$ during peak events.

In the last decade in the Barents Sea first and then in the Eurasian Arctic Ocean, ice-free regions have been observed along the path of the Atlantic Water, and warm and saline water has been extending up to the surface (Årthun et al., 2012; Ivanov

et al., 2016). The lack of sea ice is mainly due to heat from the Atlantic layer reaching the surface, and is associated with the Atlantification of the Eurasian Basin and of the Barents Sea (Polyakov et al., 2017; Årthun et al., 2012). In the Eurasian Basin, the upward oceanic heat flux towards the mixed layer has increased from $3 - 4$ W m$^{-2}$ in 2007 - 2008 to more than 10 W m$^{-2}$ in 2016 - 2018 (Polyakov et al., 2020). This process is called the ice-ocean heat feedback as the increased ocean heat flux to the sea surface reduces ice thickness and increases its mobility, increasing atmospheric momentum flux into the

ocean and reducing the damping of surface-intensified baroclinic tides (Polyakov et al., 2020). Mixing north of Svalbard is of


**Table 1.** Overview of ocean microstructure measurements. The number of profiles used in analyses is $n$, after batch-averaging repeat profiles in the process stations.

| Start | End | Instrument | Number of profiles | $n$ |
|---|---|---|---|---|
| 30 Jun 2018 17:30 UTC | 08 Jul 2018 20:00 UTC | VMP 2000 | 185 | 76 |
| 16 Sep 2018 21:30 UTC | 20 Sep 2018 04:40 UTC | VMP 2000 | 43 | 14 |

particular interest to understand the Atlantification as it contributes to the cooling and freshening of the Atlantic Water entering the Arctic Ocean. The reduced ice cover over the continental slope north of Svalbard can be seen as a precursor of the entire Eurasian Basin and the processes therein. Indeed, Polyakov et al. (2020) documented an eastward lateral propagation of the so-called Atlantification, with a lag of about 2 years between the Barents Sea and the eastern Eurasian Basin. Therefore, detailed observations of the ocean dynamics north of Svalbard are needed to evaluate the active processes modifying the Atlantic Water layer in a changing Arctic, and their potential influence on the sea ice.

In this study we present observations of ocean turbulence north of Svalbard collected in summer and fall 2018, and focus on mechanisms which lead to turbulence in the different layers of the water column. Two main sources of ocean mixing are investigated: the wind and the tidal forcing. Turbulence production by background shear will not be addressed in this study as the vertical resolution (8 m) of the current data collected during the cruises is not sufficient to resolve shear instabilities.

## 2 Data and methods

Data were collected in 2018, during two cruises that took place north of Svalbard as a part of the Nansen Legacy Project. The summer cruise was on *R/V Kristine Bonnevie* from 27 June to 10 July 2018 (Fer et al., 2019), while the fall cruise was on the ice-class *R/V Kronprins Haakon* from 12 to 24 September 2018 (Fer et al., 2020a). During the cruises, several sections were repeated north of Svalbard across the continental slope, and 3 stations (two in July and one in September) were occupied for about 24 h to study mixing processes (Figure 1b). Turbulence profiles were collected during both cruises (185 profiles in 9 days in July and 43 in 5 days in September) using a Vertical Microstructure Profiler (VMP). We use the International Thermodynamic Equations of Seawater (TEOS-10) (McDougall and Barker, 2011) with Conservative Temperature ($\Theta$) and Absolute Salinity ($S_A$).

### 2.1 Vertical Microstructure Profiler (VMP)

We used a 2000 m-rated VMP manufactured by Rockland Scientific, Canada (RSI). The VMP is a loosely tethered profiler with a nominal fall speed of $0.6\,\mathrm{m\,s^{-1}}$. The profiler was equipped with pumped Sea-Bird Scientific (SBE) conductivity and temperature sensors, a pressure sensor, airfoil velocity shear probes, one high-resolution temperature sensor, one high-resolution micro-conductivity sensor and three orthogonal accelerometers. The microstructure data were processed using the routines





provided by RSI (ODAS v4.01). Assuming isotropic turbulence, the dissipation rate of turbulent kinetic energy per unit mass, $\epsilon$, can be expressed as

$$\epsilon = 7.5\nu \overline{\left(\frac{\partial u}{\partial z}\right)^2} \tag{1}$$

where $\nu$ is the kinematic viscosity equal to about $1.6 \times 10^{-6}$ m$^2$ s$^{-1}$ in these temperatures, overbar denotes averaging in time and $\partial u/\partial z$ is the small-scale shear of one horizontal velocity component $u$. Dissipation rates were calculated from the shear
variance obtained by integrating the shear vertical wavenumber spectra in a wavenumber range that is relatively unaffected by noise, and corrected for the variance in the unresolved portions of the spectrum using an empirical model (Nasmyth, 1970). The shear spectra were computed using 1 s Fourier transform length and half-overlapping 4 s segments. We quality screened the resulting values by inspecting the instrument accelerometer records, individual spectra and individual dissipation rate profiles from the two shear probes. We averaged estimates from both probes, except when their ratio exceeded 10, for example as
a result of plankton hitting a sensor, the lowest estimate was chosen. Noise level of the dissipation rate measured by the VMP is about $(2-3) \times 10^{-10}$ W kg$^{-1}$. The temperature and salinity data from the VMP were compared against the ship's SBE CTD profiles. A good agreement was observed and no correction was made. In total, we collected 31 profiles. Dissipation measurements from the upper 15 m were excluded because of the disturbance from the ship's keel, and the profiler's adjustment to free fall. The vertically integrated dissipation rate over a layer $h$ (surface mixed layer or near-bottom layer in the following
sections) is defined as $D_h = \rho_0 \int_h \epsilon(z)dz$ (in W m$^{-2}$) where $\rho_0 = 1027$ kg m$^{-3}$ is the seawater reference density.

We estimated the turbulent heat flux $F_H$ from

$$F_H = -\rho_0 C_p \kappa \frac{\partial \Theta}{\partial z}, \tag{2}$$

where $C_p = 3991.9$ J kg$^{-1}$ K$^{-1}$ is the specific heat of seawater, $\Theta$ is the background temperature and $\kappa$ is the diapycnal eddy diffusivity. We thus assume that turbulence diffuses the finescale temperature gradient at the same rate as the density gradient.
The sign convention is that positive heat fluxes are directed upward in the water column.

We expressed the diapycnal diffusivity $\kappa$ as a function of turbulent activity index, following Bouffard and Boegman (2013), where three states (energetic, transitional and buoyancy-controlled) are defined depending on the Reynolds number $Re_b = \frac{\epsilon}{\nu N^2}$. To compute $\kappa$, the buoyancy frequency or Brunt Väisälä frequency, $N$, was calculated using $N^2 = -\frac{g}{\rho_0} \frac{\partial \sigma_0}{\partial z}$, where $g$ is the gravitational acceleration and $\sigma_0$ is the potential density anomaly referenced to surface pressure. Background vertical gradients
(for temperature, salinity and density) were taken over a 10-m length scale. As $N$ approaches neutral stratification, $\kappa$ attains very large values. The estimates of $\kappa$ in segments with buoyancy frequency below a noise level of $N^2 = 10^{-7}$ s$^{-2}$ were excluded.

We also computed the salinity flux $F_S$ and the buoyancy flux $F_B$, with the same sign convention as the turbulent heat flux:

$$F_S = -\rho_0 \kappa \frac{\partial S_A}{\partial z}, \tag{3}$$






$$F_B = -g(\beta F_S - \alpha F_H),\tag{4}$$

where $\alpha$ and $\beta$ are respectively the thermal expansion and salinity contraction coefficients.

## 2.2   Other datasets

We used the profiles collected from the ship's CTD system (Sea-Bird Scientific, SBE 911plus on both cruises) to check and
correct the temperature and salinity from the VMP. CTD data were processed using the standard SBE post-processing software, and salinity values were corrected against water sample analyses. Pressure, temperature and practical salinity data are accurate to $\pm 0.5$ dbar, $\pm 2 \times 10^{-3}$ °C, and $\pm 3 \times 10^{-3}$, respectively.

The wind speed, direction and surface air temperature (Figure 2) were recorded every minute during the cruises from the ship's weather station.

We used Arc5km2018 (Erofeeva and Egbert, 2020), a barotropic inverse tidal model on a 5-km grid, to estimate the tidal currents using the 8 main constituents ($M_2$, $S_2$, $N_2$, $K_2$, $K_1$, $O_1$, $P_1$, $Q_1$) and 4 nonlinear components ($M_4$, $MS_4$, $MN_2$, and $2N_2$).

Bathymetric contours shown in maps are from the International Bathymetric Chart of the Arctic Ocean (IBCAO-v3) (Jakobsson et al., 2012). Station depths are from the ship's echosounder.

To discuss our findings in a broader scope, we used the global monthly isopycnal mixed-layer ocean climatology (MIMOC) at $0.5°$ resolution, which is objectively mapped with emphasis on data from the last decade (Schmidtko et al., 2013).

## 3   Overview of observations

In the rest of the study, sets of 4-5 consecutive repeat profiles at the process stations are averaged to avoid any bias toward these stations. Table 1 lists two numbers of 'profiles': the total number of casts performed (number of profiles) and the number
of profiles used in analyses ($n$) after batch-averaging of consecutive repeat profiles. In the rest of the study, we always refer to the number of profiles after batch-averaging (in figures 4 and 8).

### 3.1   Environmental context

The cruises cover the summer and fall conditions, typically in open waters. Four main sections were occupied north of Svalbard: Section B, C and E in July, and section D in September, capturing the core of the inflowing Atlantic Water. Selected stations
were occupied for 24 hours to investigate mixing processes in detail at a specific location: T, RS1, RS2 and P (Figure 1b). In September, turbulence profiling terminated after the winch broke, resulting in fewer profiles (section D, process study P and the outer deep stations at section C).

In July, the Yermak Plateau was covered by sea ice and the ice edge was close to the continental slope north of Svalbard (Figure 1), limiting the station coverage (e.g., section D could not be completed). We note that the sea ice encounter in July was



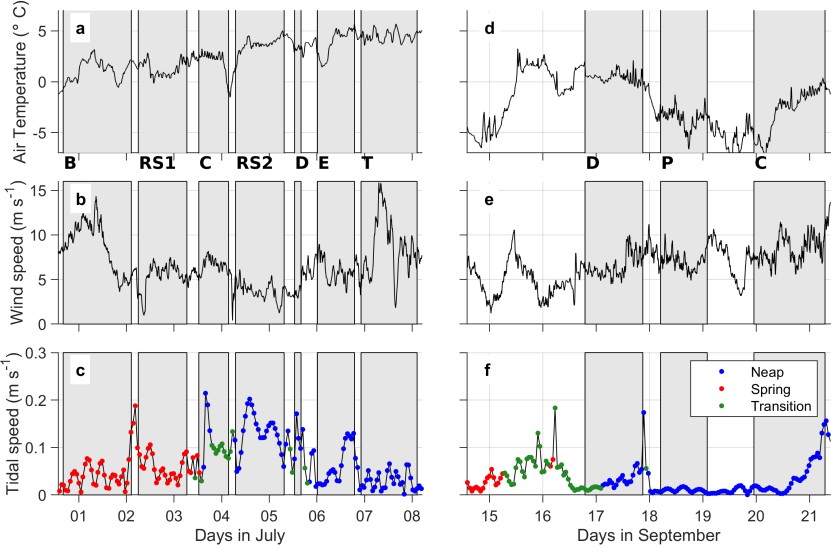

**Figure 2.** Air temperature (a and d), wind speed (b and e) from the ship's weather station and tidal current speed (c and f) from the Arc5km2018 model. Left panels are during the July (Summer) cruise. Right panels are during the September (fall) cruise. Grey shadings correspond to the periods of turbulence measurements (sections or process stations). In panel c and f, the tidal conditions during time of sampling are indicated as neap, spring tide and the transition between the neap and spring tide with blue/red/green dots respectively.

closer to the continental slope at 24°E than what is suggested by the sea ice edge from satellite. In September, the sea ice edge was ∼ 30 to 50 km further north, and the continental slope was entirely free of ice. Open water conditions on shelves were previously observed to facilitate enhanced wind energy input to the oceanic near-inertial currents (Rainville and Woodgate, 2009). The momentum transfer from the wind is also affected by ice conditions in deep basins. The drifting Ice-Tethered Profiler dataset in the Canada Basin from 2005 to 2014 shows that the near-inertial internal wave field is the most energetic in

summer when sea ice is at a minimum (Dosser and Rainville, 2016).

Air temperature differs between the two cruises: while it was mainly positive in July, the temperature dropped to -10°C in September (Figure 2a and d) near the sea ice edge. Over the two cruises, wind was moderate, peaking only for half a day to 15 m s$^{-1}$ on 7 July (Figure 2b). In September, the average wind speed was 8 m s$^{-1}$ with no specific events.

Tidal currents varied significantly during the cruise depending on the location, with a maximum amplitude about 20 cm s$^{-1}$

during RS2 from 4 to 6 July. The tidal currents were stronger on the slope than in the deep basin (such as during P in the Nansen Basin). In July, sections were occupied during spring tides for the first 3 days and during neap tides for the rest of the cruise. In September, the tidal currents at the stations were weaker and mainly during neap tides, except for a short period of spring tides in the beginning of the cruise (around 15 September 2018).





## 3.2 Hydrography

Figure 3 shows the distribution of temperature and dissipation rate collected in sections and the process stations performed during the two cruises. Temperature sections were obtained by gridding the data in 1 km horizontal and 2 m vertical grid size. Note that in Figure 3, the horizontal axis in the left panels (sections) is distance to the 800 m isobath (negative onshore and positive offshore), and time in the right panels (process stations).

We estimate the mixed layer depth (red line in Figure 3), as the depth at which the density exceeds the shallowest mea-
surement by 0.01 $kg\,m^{-3}$ in July, and by 0.03 $kg\,m^{-3}$ in September, because of the presence of melt water at the surface in September. The vertical gradients are large, and the mixed layer depth is not very sensitive to the exact criteria. We also estimate a surface layer depth following Randelhoff et al. (2017):

$$r(z) = \frac{\sigma_0(z) - \sigma_{0s}}{\Delta\sigma_0} \qquad (5)$$

where $\Delta\sigma_0 = \sigma_{0d} - \sigma_{0s}$ is the surface density deviation, $\sigma_{0d}$ is the density averaged between 55 and 65 m and $\sigma_{0s}$ is the
density averaged between 5 and 8 m. The surface layer depth is useful when there is no well-defined mixed layer but a seasonal pycnocline. In our dataset, $\Delta\sigma_0$ is typically larger than 0.02 $kg\,m^{-3}$, hence the surface layer is discernibly affected by meltwater (Randelhoff et al., 2017); however the surface density deviation is larger in September: the meltwater layer is deeper and fresher in fall than in summer. Here, the mixed layer depth and the surface layer depth are very similar (not shown). In the rest of the study, we use the mixed layer depth using the density difference criterion (red line in Figure 3).

The slope north of Svalbard is characterized by Atlantic Water flowing along the 800 m isobath. The Atlantic Water is defined as water masses with $\Theta > 2°C$ and $27.7 < \sigma_0 < 27.97$ $kg\,m^{-3}$ following Rudels et al. (2000). The warm waters observed roughly between 500 and 1100 m isobaths are associated with the Atlantic Water core (section panels in figure 3). Colder and fresher waters found offshore are Atlantic Water from Fram Strait, which has been modified by mixing with the surrounding waters. A thorough description of the hydrography and circulation during the two cruises can be found in Kolås
et al. (submitted).

We calculated average profiles of temperature, salinity, dissipation rate and diffusivity in isopycnal coordinates, to account for the possible vertical displacement of the isopycnals and of the water masses from the slope to the deep basin. Once averaged, the profiles are mapped into vertical coordinate using the corresponding average depth of an isopycnal (Figure 4). The average profiles are obtained in subsets, depending on their distance from the 800 m isobath, which is representative of the mean
location of the core of the inflowing Atlantic Water (Kolås et al., submitted). The core of the Atlantic Water current typically extends about 20 km onshore and offshore of the 800 m isobath (Kolås et al., submitted). However, in order to characterize the different regions of the slope with comparable number of profiles in each region, we present averages inshore of -10 km, within ±10 km of the 800 m isobath and offshore of 10 km (Figure 4).

Averaged temperature and salinity profiles are very similar at depth (below 600 m, around 0° C and 35.1 g $kg^{-1}$, Figure
4e), and the main differences are observed in the upper 200 m. The 'inshore' average profile is the warmest with a temperature



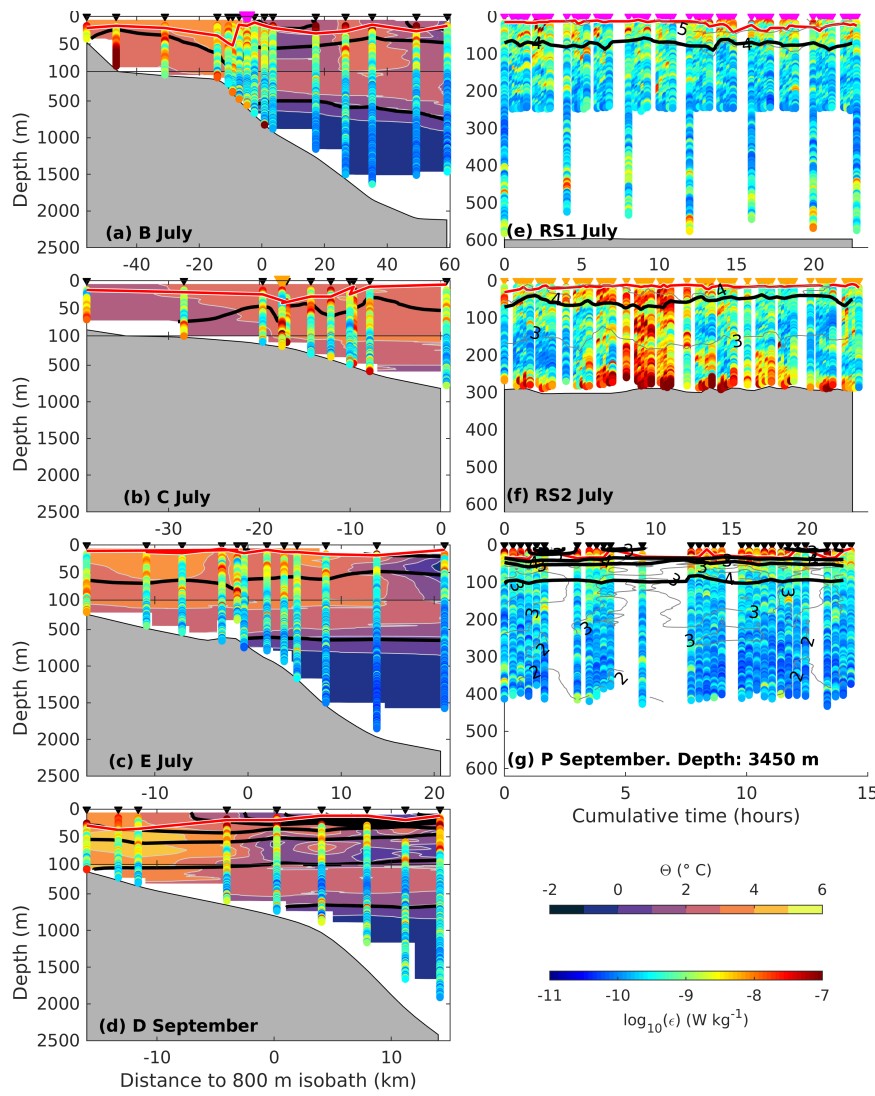

**Figure 3.** Overview of the main sections (left panels) and of the process studies (right panels) during the July and September cruises. In the left panels, background is Conservative Temperature, superimposed are the dissipation rate profiles ($\epsilon$). Note the change of vertical scale at 100 m depth. In the right panels, temperature contours as shown as thin grey lines. In all panels, bold black lines are the isopycnals. Bathymetry is from the bottom depth measured at each station. Triangle markers are the time/location of the stations. In panels a and b, the pink and orange station markers indicate the location of the RS1 and RS2 process station, respectively, shown in panels e and f. At station P (g), one early VMP cast performed about 6 h before the start of the first shown profile is excluded. The horizontal axis is the distance to the 800 m isobath in the left panels and cumulative time from the first profile in the right panels.

maximum of $\sim 5.5°$C at around 75 m depth. The 'offshore' average profile has the coldest mixed layer (around $0°$C) and the coldest core of Atlantic Water (around $2°$C), a characteristic of the hydrography in the Nansen Basin (Kolås et al., submitted).


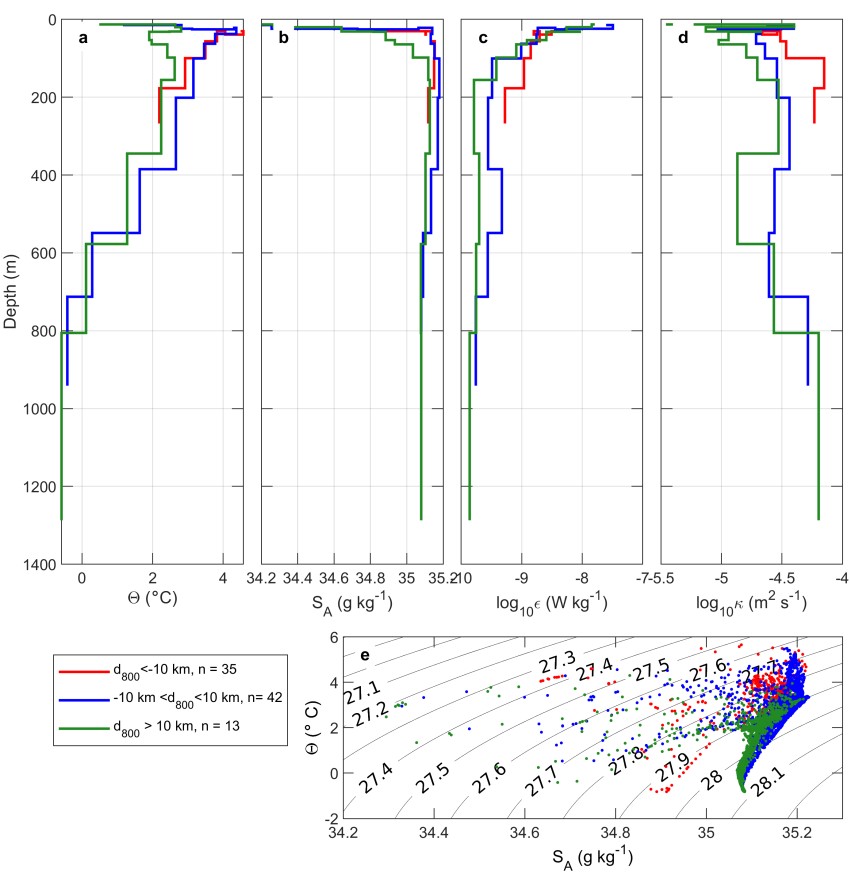

**Figure 4.** Isopycnally-averaged profiles of a) $\Theta$, b) $S_A$, c) dissipation rate, $\epsilon$ and d) diapycnal diffusivity, $\kappa$. The profiles are shown using the average depth of the isopycnals. e) Temperature-Salinity diagram. Profiles are selected relative to distance from the 800-m isobath: inshore of -10 km (red), at the shelf break in the Atlantic Water core (blue), and offshore of 10 km (green). In the legend, $n$ indicates the number of batch-averaged profiles used in each average.

### 3.3 Turbulence

On average, dissipation rates are the largest in the upper ocean, reaching $10^{-7}$ W kg$^{-1}$ near the surface and decreasing rapidly
with depth (Figure 4c). In deeper layers, the dissipation rates are larger inshore than offshore, decreasing from $5 \times 10^{-10}$ W kg$^{-1}$ on the shallows (red profiles) to $10^{-10}$ W kg$^{-1}$ on the deep offshore profiles (green profiles). Between 400 and 600 m depth, a local maximum in dissipation rate is observed in the core of the inflowing Atlantic Water current (blue profiles), where the strongest currents are observed. Diffusivity is large in both the mixed layer and at depth close to the bottom (Figure 4d), exceeding $6 \times 10^{-5}$ m$^2$ s$^{-1}$.





Of the microstructure measurements collected during the cruises, the process stations RS2 and P were analyzed and reported
in detail in Fer et al. (2020b) and Koenig et al. (2020), respectively. The largest dissipation rates were measured at RS2, with
high dissipation rates observed in the whole water column during a 6-h turbulent event (Figure 3f), caused by an intense
dissipation of lee waves driven by cross-slope tidal currents (Fer et al., 2020b). Process station P in the Nansen Basin far from
the continental slope (Figure 1) is a 24 h process study at a surface thermohaline front (Figure 3g). At this specific station,

turbulence structure in the mixed layer was generally consistent with turbulence production through convection by heat loss to
atmosphere and mechanical forcing by moderate wind (Koenig et al., 2020).

In the following sections, we will first examine the mixed layer evolution from summer to fall. Then we will investigate
the sources of turbulence at depth within and below the Atlantic Water layer. Basic statistics (arithmetic and geometric mean
and standard deviations) of mixing parameters for July and September are summarized in Table 2. We used both arithmetic

and geometric mean to describe the dissipation rates, diffusivity and turbulent fluxes. For variables with lognormal distribution
(such as $\epsilon$ and $\kappa$), the geometric mean (GM) characterizes the distribution's central tendency while the arithmetic mean (AM)
tends to be disproportionately skewed by a small number of large values (Scheifele et al., 2020). AM characterizes the inte-
grated effect of the distribution and is representative of the cumulative effect of mixing and average buoyancy transformations
produced by mixing (Scheifele et al., 2020).

## 4   Upper layer dynamics

### 4.1   Seasonal evolution

Solar heating melts the sea ice which has consequences for the upper ocean dynamics. Throughout the summer, the mixed
layer becomes fresher and lighter (34.9 g kg$^{-1}$ and 27.7 kg m$^{-3}$ in July, and 34 g kg$^{-1}$ and 26.95 kg m$^{-3}$ in September, figure
5), and also deepens (18 m in July and 23 m in September). This evolution in summer towards a lighter mixed layer is mainly

due to the meltwater during the summer. In both summer and fall, dissipation rates, buoyancy fluxes and turbulent heat fluxes
increased at the base and just below the mixed layer compared to the rest of the water column (figure 5 and table 2).

The depth-integrated dissipation rate at the base of the mixed layer $D_{ml}$ is on average about $(1-2) \times 10^{-4}$ W m$^{-2}$ in
both cruises with a geometric mean of about $3 \times 10^{-5}$ W m$^{-2}$ (table 2). In both July and September downward turbulent
buoyancy fluxes and upward turbulent salinity fluxes are observed in the mixed layer. Salinity and buoyancy fluxes are larger

in September than in July at the base of the mixed layer: salinity flux is $4.2 \times 10^{-4}$ W kg$^{-1}$ in July and $1.1 \times 10^{-3}$ W kg$^{-1}$ in
September and buoyancy flux is $-2.4 \times 10^{-9}$ W kg$^{-1}$ in July and $-5.3 \times 10^{-9}$ W kg$^{-1}$ in September, as the meltwater content
in the upper layer is larger in September than in July.

Turbulent heat fluxes are positive (upward) in both July and September, but larger in September than in July (3.6 W m$^{-2}$ and
2.5 W m$^{-2}$ respectively). The turbulent heat fluxes measured during both cruises are comparable to what is observed under

the sea ice without any specific forcing during the N-ICE2015 experiment (Meyer et al., 2017; Peterson et al., 2017) (about 2
W m$^{-2}$), but about 1/40 to 1/30 times the heat fluxes (up to 100 W m$^{-2}$) observed during storm events above the continental





**Table 2.** Statistics of the turbulence variables measured in July and September. AM: arithmetic mean, GM: geometric mean, $\sigma$: standard deviation, $D_\epsilon$: vertically integrated dissipation rate, $\epsilon$: dissipation rate, $\kappa$: diffusivity, $F_H$: vertical turbulent heat flux (positive upward). Four layers are defined. MLD: $\pm 10$ m around the base of the mixed layer, $AW_{core}$ - MLD: from the Atlantic Water core to the mixed layer depth, AW layer: in the Atlantic Water layer, and $AW_{core}$ - bottom: from the Atlantic Water core to the seafloor. The geometric mean is ill-defined for negative values hence not provided for the turbulent heat fluxes.

| | | July | | | Sept | | |
|---|---|---|---|---|---|---|---|
| | | AM | GM | $\sigma$ | AM | GM | $\sigma$ |
| | MLD | 1.3 | 0.3 | 2.5 | 1.8 | 0.3 | 4.6 |
| $D_\epsilon \times 10^{-4}$ (W m$^{-2}$) | $AW_{core}$ - MLD | 3.1 | 0.6 | 7.2 | 3.8 | 0.7 | 8.3 |
| | AW layer | 8.9 | 5.0 | 12.8 | 8.7 | 4.8 | 12.8 |
| | $AW_{core}$ - bottom | 9.3 | 6.2 | 12 | 9.6 | 6.4 | 12.0 |
| | MLD | 23.7 | 5.3 | 56.5 | 28.4 | 5.5 | 67.6 |
| $\epsilon \times 10^{-9}$ (W kg$^{-1}$) | $AW_{core}$ - MLD | 18.8 | 4.4 | 50.1 | 22.6 | 4.8 | 55.5 |
| | AW layer | 4.1 | 1.7 | 10.3 | 4 | 1.7 | 10.3 |
| | $AW_{core}$ - bottom | 3.9 | 1.5 | 10.3 | 3.9 | 1.6 | 10.3 |
| | MLD | 6.9 | 3.9 | 6.9 | 38 | 4 | 221 |
| $\kappa \times 10^{-5}$ (m$^2$ s$^{-1}$) | $AW_{core}$ - MLD | 5.4 | 2.6 | 7.7 | 16.6 | 3.0 | 64.8 |
| | AW layer | 10.9 | 7.6 | 13.3 | 9.8 | 6.5 | 13.2 |
| | $AW_{core}$ - bottom | 11.1 | 7.9 | 13.3 | 10.5 | 7.4 | 13.1 |
| | MLD | 2.5 | X | 18.4 | 3.6 | X | 17.6 |
| $F_H$ (W m$^{-2}$) | $AW_{core}$ - MLD | 4.4 | X | 11.4 | 3.0 | X | 8.9 |
| | AW layer | -1.4 | X | 1.6 | -1.4 | X | 1.7 |
| | $AW_{core}$ - bottom | -1.5 | X | 1.6 | -1.4 | X | 1.6 |

slope. Variations in the density field in the Arctic are dominated by the variations in salinity, thus buoyancy and salinity fluxes vary concomitantly.

## 4.2 Wind forcing

Wind stress at the ocean surface is one of the main drivers for the upper layer turbulence and can increase the ocean-to-ice heat fluxes (Meyer et al., 2017; Dosser and Rainville, 2016). The wind energy flux from the atmosphere into the ocean can be estimated from the wind speed at 10 m height ($U_{10}$) as: $E_{10} = \tau U_{10} = \rho_{air} C_d U_{10}^3$ (Oakey and Elliott, 1982), where $\rho_{air}$ is the density of air and $\tau$ is the wind stress, parameterized using a quadratic drag with a drag coefficient $C_d$. We use the neutral drag coefficient at 10 m computed following Large and Pond (1981), adjusting the wind speed measured at 15 m height in July

and 22 m height in September from the ship's mast to 10 m. A fraction of this energy flux then fuels turbulence in the upper ocean and is dissipated in the mixed layer. Using the observed dissipation in the mixed layer and the wind energy input north



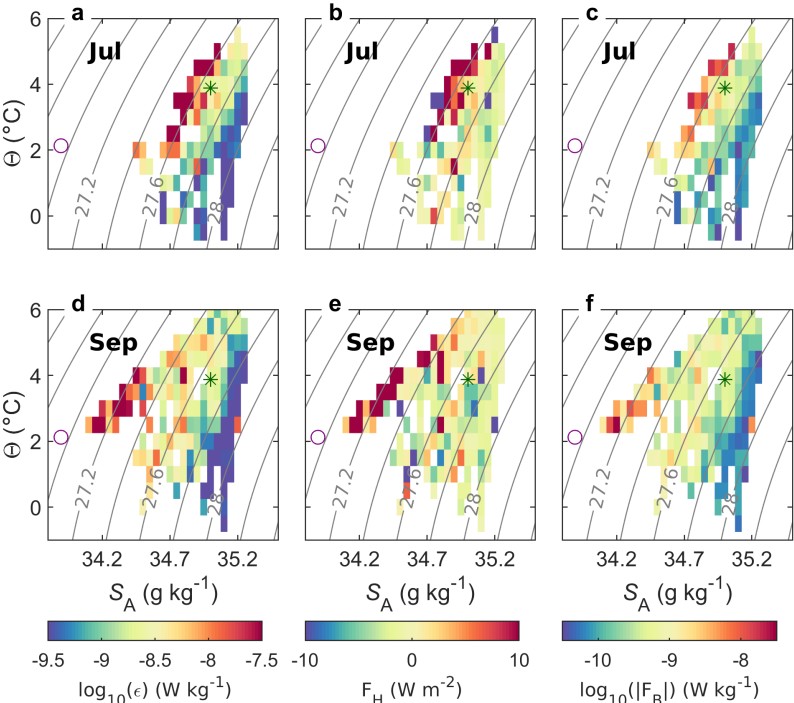

**Figure 5.** Temperature-Salinity diagrams where the color-coded bins are dissipation rate (a and d), turbulent heat flux (b and e) and magnitude of buoyancy flux (c and f, the buoyancy fluxes are all oriented downward). Contours are $\sigma_0$, referenced to surface pressure. Panels a, b and c are for summer (July cruise) and panels d, e and f for fall (September cruise). The green star is the mean temperature and salinity property of the mixed layer in July and the purple circle is the corresponding value in September.

of Svalbard (Figure 6), we obtain: $D_{ml} = 0.002 E_{10}^{(1.4\pm0.2)}$. Observations in September are limited as only a few stations were performed with the VMP.

For relatively low values of $E_{10}$ (less than $6.3 \times 10^{-1}$ W m$^{-2}$), the relation is almost linear, suggesting that about 1 per mille

of wind energy input is dissipated in the mixed layer. For larger $E_{10}$, additional processes such as breaking gravity waves can contribute. The front process station P is more energetic than what is expected from only wind forcing as convection is active on the warm side of the front (green star) (Koenig et al., 2020). Dissipation in the mixed layer at RS2 (orange circle) is only computed using the first casts as there is no data in the shallow mixed layer during the intense dissipation event driven by cross-slope tidal currents (Figure 3f). The presence of sea ice in the region can also explain the non-linearity of the relation between

the wind energy and the energy dissipation in the mixed layer. Although the profiles were collected in ice-free conditions, some stations were close to the sea ice edge spotted from the ship.

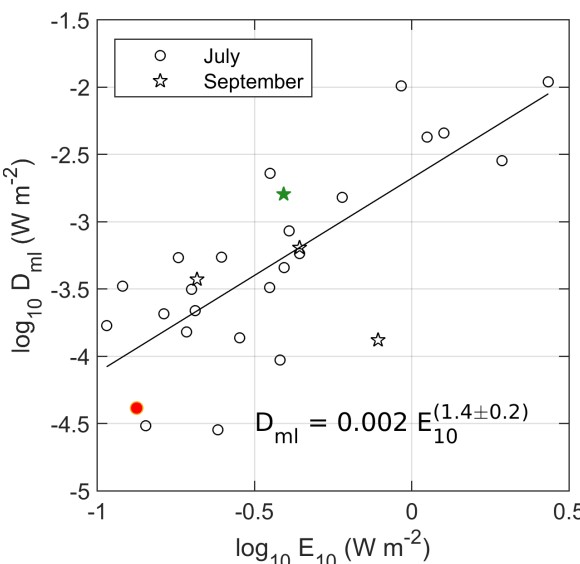

**Figure 6.** Depth-integrated dissipation rate in the mixed layer, $D_{ml}$, as a function of the wind energy input to the mixed layer, $E_{10}$. Stars and circles are data from the September and July cruises, respectively. The red circle is the data point from the RS2 process station where nonlinear internal waves were observed (Fer et al., 2020b). The green star is the data point at the process station at a front in September where convection was also important (Koenig et al., 2020). The black line is the regression line, with the equation indicated. The uncertainty is the 95% confidence interval.

## 5 Mixing in the Atlantic Water layer

As the heat content in the Atlantic Water in the Arctic Ocean has the potential to melt the sea ice cover completely, it is important to quantify the turbulent dissipation rates and heat fluxes out of the Atlantic Water in present conditions of a warming Arctic.

Depth-integrated dissipation rate from the base of the mixed layer to the Atlantic Water core is about $8.8 \times 10^{-4}$ W m$^{-2}$, and the average dissipation rate is about $2 \times 10^{-8}$ W kg$^{-1}$, almost as large as what is observed in the mixed layer (table 2). We estimated the vertical turbulent heat flux between the upper limit of the Atlantic Water layer and the mixed layer depth (Figure 7a), in both summer and fall. Maximum positive heat flux (upward toward the surface) is observed near the 800 m isobath, reaching up to 30 W m$^{-2}$ in July and 10 W m$^{-2}$ in September. This isobath is representative of the average location

of the core of Atlantic Water (Kolås et al., submitted). Outside the Atlantic Water boundary current, at about 20 km inshore and offshore from the 800 m isobath, vertical turbulent heat fluxes are negligible, with a maximum of 5 W m$^{-2}$. In July, the Atlantic Water core tends to be closer to the base of the mixed layer compared to September (Figure 7 a), implying that the heat contained in the Atlantic Water is more likely to reach the surface in July than in September. Meltwater in September enhances the stratification near the surface and isolates the Atlantic Water layer from the mixed layer. At some stations, vertical turbulent

heat fluxes are negative (less than 5 W m$^{-2}$), directed downward from the surface toward the Atlantic Water layer. The negative



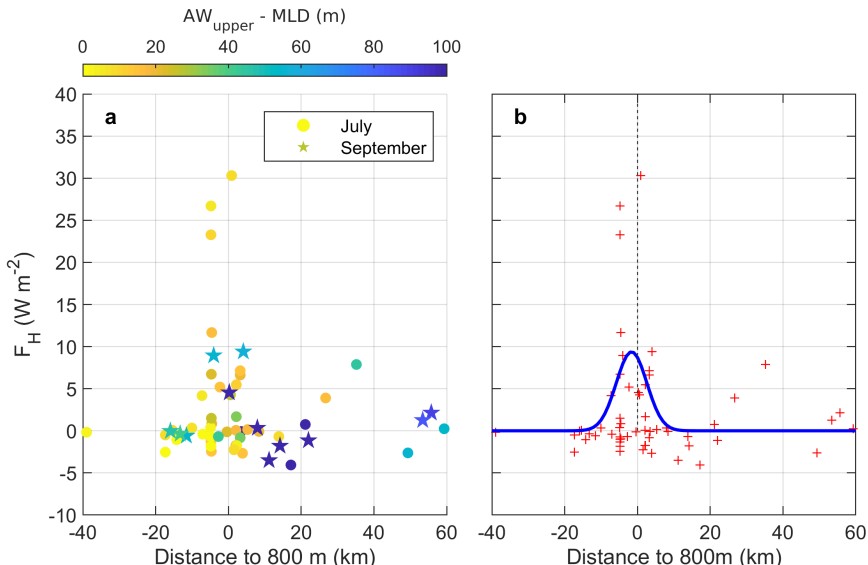

**Figure 7.** a) Lateral distribution of the mean vertical turbulent heat flux from the AW upper boundary to the base of the mixed layer. Horizontal axis is the horizontal distance to the 800 m isobath. Color code is the vertical distance between the upper boundary of the Atlantic Water layer and the mixed layer depth. Markers identify the stations collected in July (circles) and September (stars). b) A Gaussian fit (blue line) to the vertical turbulent heat flux from the Atlantic Water to the mixed layer depth (red crosses).

fluxes are mainly found near the core of the Atlantic Water inflow. These negative heat fluxes are observed when warm water reaches the surface, and the temperature increases from the top of the Atlantic Water layer up to the surface. This situation is typical of summer conditions north of Svalbard where the Atlantic Water extends close to the surface and the cold halocline is absent (Polyakov et al., 2017).

The lateral (cross-isobath) distribution of the diapycnal heat fluxes is similar in July and September (Figure 7a). We therefore used all data points to fit a Gaussian curve (Figure 7b), with an aim to estimate the integrated heat loss from the Atlantic Water layer. Between -20 and +20 km, the heat loss due to vertical turbulent heat fluxes is about $1.2 \times 10^5$ W m$^{-1}$ across the section. Using independent hydrographic observations but covering the same observational time period, Kolås et al. (submitted) found that the average along-path change of heat content from section B to E was about $9.1 \times 10^7$ W m$^{-1}$, and about $9.6 \times 10^6$ W m$^{-1}$

from section C to D, corresponding to an average heat loss of about 500 W m$^{-2}$ north of Svalbard. Heat loss from the Atlantic Water layer by vertical turbulent heat fluxes to the upper ocean then accounts for only about 1% of the total Atlantic Water heat loss north of Svalbard. This estimate can be biased low since during the period of measurements wind forcing was weak to moderate with low variability. Processes that contribute to the turbulent heat loss of the Atlantic Water layer are discussed in section 7.





## 6  Tidal mixing

Previous observations show that north of Svalbard is a region of substantial tidal mixing (Rippeth et al., 2015; Fer et al., 2014). The location is northward of the critical latitude of the main diurnal and semi-diurnal tidal components ($M_2$ and $K_1$). The critical latitude, also called the turning latitude, is where the tidal frequency matches the local inertial period. The linear response at high latitudes is evanescent. The barotropic to baroclinic energy conversion from the tidal activity results in trapped linear waves that can only propagate along topography guidelines, or nonlinear response with properties similar to lee waves (Vlasenko et al., 2003; Musgrave et al., 2016). A fraction of the energy in trapped waves or nonlinear waves will dissipate locally, leading to substantial vertical mixing (Padman and Dillon, 1991). In our observations, the dissipation rate below the mixed layer is typically low (table 2), but energetic turbulence observed at some locations (Figure 3) can be related to tidal forcing.

We select the profiles of turbulent heat fluxes and dissipation rates in categories of tidal current speed predicted from Arc5km 2018 at the time of the measurement. Tidal current speed is defined as large ($> 5 \, \mathrm{cm \, s^{-1}}$) or low ($< 5 \, \mathrm{cm \, s^{-1}}$) (Figure 8). The profiles in the corresponding categories are averaged with respect to height above bottom defined as the difference between the depth of the measurement and the seafloor depth. We obtained the average profiles as the maximum likelihood estimator from a lognormal distribution using the data points in 20 m vertical bins. The mixed layer was excluded in all the profiles to minimize the contribution from dissipation driven by surface processes.

From the seafloor to about 250 m height above bottom, dissipation rate was larger ($\epsilon > 10^{-8} \, \mathrm{W \, kg^{-1}}$) in conditions with strong tidal currents compared to weaker tidal currents ($\epsilon < 5 \times 10^{-9} \, \mathrm{W \, kg^{-1}}$). In both cases, the dissipation rate decreases quickly with height from the seafloor, down to dissipation rates of $\sim 5 \times 10^{-10} \, \mathrm{W \, kg^{-1}}$ above 250 m from the bottom. Increase in dissipation rates for strong tidal forcing is associated with an absolute increase in the downward turbulent heat flux close to seafloor: -2.2 W m$^{-2}$ when tidal currents were weak and about -3.2 W m$^{-2}$ when tidal currents were strong.

Similar to the dissipation rate, the diapycnal diffusivity decreases with increasing height above bottom (Figure 8c). Based on the observations from north of Svalbard, we can deduce an empirical relation that would allow an estimate of the diffusivity in conditions of strong ($> 5 \, \mathrm{cm \, s^{-1}}$) or weak ($< 5 \, \mathrm{cm \, s^{-1}}$) tidal currents. Following St. Laurent et al. (2002), we use a functional form for the diffusivity expressed as:

$$\kappa = \kappa_{bg} + \kappa_{bot} \times e^{-h/z_{decay}}, \tag{6}$$

where $\kappa_{bg}$ is a background diffusivity, $\kappa_{bot}$ is the diffusivity value at the seafloor, $h$ is the height above bottom and $z_{decay}$ is the vertical decay scale of the diffusivity. We use $\kappa_{bot} = 5 \times 10^{-5} \, \mathrm{m^2 \, s^{-1}}$, based on observations. Fitted equations for large and low tidal current amplitude are shown above Figure 8c. With large tidal amplitudes, $\kappa_{bot}$ approximately doubles and the decay scale increases from 18 to 22 m.

We investigate the role of two distinct contributions from tidal currents. While tidally-driven processes may lead to interior mixing away from the sea bed (Fer et al., 2020b), bottom stress from barotropic tidal currents must be balanced by dissipation in bottom boundary layers. The tidal work can be representative of the barotropic-to-baroclinic conversion and can be related to

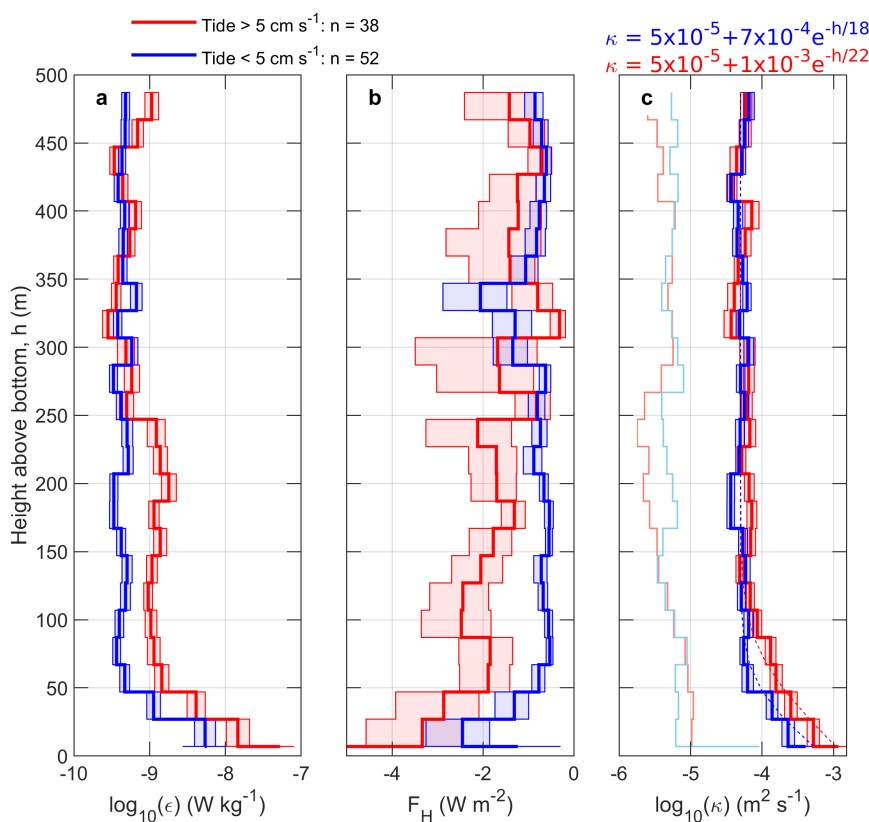

**Figure 8.** Average profiles of a) dissipation rate, b) turbulent heat flux and c) diapycnal diffusivity $\kappa$ for small (blue) and large (red) tidal current amplitudes estimated from Arc5km2018 at the time and location of each station. Average profiles are obtained as the maximum likelihood estimator from a lognormal distribution using the data points in $20\,\mathrm{m}$ vertical bins. The vertical axis is height above bottom, relative to the seafloor depth from the echo sounder. $n$ indicates the number of batch-averaged profiles for each tidal forcing category. Thin lines in (c) are the corresponding profiles for diffusivity at the lowest detection level (obtained by imposing a noise level for dissipation rate of $1 \times 10^{-10}\,\mathrm{W\,kg^{-1}}$). The dashed lines are the curve fits using an exponential function. Resulting equations with the best fit coefficients are shown above the panel. The shading is the 95% confidence envelope of the maximum likelihood.

propagating or trapped internal waves dissipation and can likely extend farther into the water column. In the bottom boundary layer, the bottom stress from the barotropic tide also plays a role. The relative contributions to mixing through the tidal work and the tidally-driven bottom drag are unknown.

We analyse the vertically integrated dissipation rate in the bottom 250 m of the water column $D_{250}$ (and below the surface mixed layer) separately with respect to tidal work and the tidally-driven bottom drag (Figure 9). Following Nash et al. (2006), the total rate of work by barotropic tidal currents interacting with topography is the product of the Baines force (Baines, 1982),





$F_{Baines} = N^2 w \omega^{-1}$, and the barotropically induced vertical velocity, $w = \mathbf{u} \cdot \nabla \mathbf{H}(z/H)$, where $\omega$ is the tidal wave frequency,

$N(z)$ is the Brunt Väisälä frequency, $\mathbf{u}$ is the tidal current vector and $\nabla \mathbf{H}$ is the bottom slope vector. Excluding the wave frequency as a variable, we introduce the following parameter related to tidal work:

$$W_{tidal} = N^2 |\mathbf{u} \cdot \nabla \mathbf{H}|^2. \tag{7}$$

where N is the Brunt Väisälä frequency near bottom (averaged in the $100\,\mathrm{m}$ height above bottom). The tidally-driven bottom drag is expressed as in Jayne and St. Laurent (2001):

$$W_{botdrag} = \rho_0 C_d |\mathbf{u}|^3, \tag{8}$$

where $C_d$ is the bottom drag coefficient, $\rho_0$ is the seawater density and $\mathbf{u}$ is the tidal current vector.

In both equations, two different tidal currents are used: the instantaneous tidal speed $u_t$ at the time and location of each station and a statistical estimate of the representative cross-isobath tidal current, $u_{rms}$, at the location of each station (Figure 9). Both are obtained from the Arc5km2018 model. We calculated $u_{rms}$ from the predicted local cross-isobath component of

the tidal currents over an arbitrary 30-day window using all constituents; this choice offers a parameter easily available for parameterization purposes, independent of observations.

In analyzing the vertically integrated dissipation rates with respect to local forcing at the time of observations (Figures 9a, c), we averaged the process stations in batches as explained in section 3; this allows for including some time variability in the observations. For the analysis of typical tidal forcing (not time variable), we averaged each process station as one data point

because each location is associated with a time independent $u_{rms}$ (Figure 9b, d). $D_{250}$ does not correlate well with the tidal work-related parameter at the time of observations (figures 9a). A least-squares power-law fit results in an uncertainty on the power which is more than 30% of its estimated value. The local bottom drag at the time of observations correlates somewhat better with $D_{250}$, and follows the power-law fit with a considerable scatter (uncertainty on the power is reduced to less than 25% its estimated value, figures 9c). If we force a linear relation in panel c, we find a drag coefficient of $C_d = 8.2 \times 10^{-4}$. This value

is of the same order of magnitude as the one deduced from in situ observations in the Bering Strait: $2.3 \times 10^{-3}$ (Couto et al., 2020). The red data points in figure 9 are from the station RS2, where large mixing was observed from dissipation of non-linear internal waves (Fer et al., 2020b). It partly explains why larger dissipation rates are observed here than what could be expected from the tidally-driven bottom drag alone. The analysis is repeated using the tidal work and bottom drag parameters calculated using the typical cross-isobath tidal forcing $u_{rms}$ (Figure 9b and d). The scatter is reduced, and particularly, the bottom-drag

relation offers a useful parameterization to infer vertically-integrated dissipation rates. A pan-Arctic coarse estimate will be given in the following section using the relation in Figure 9d.

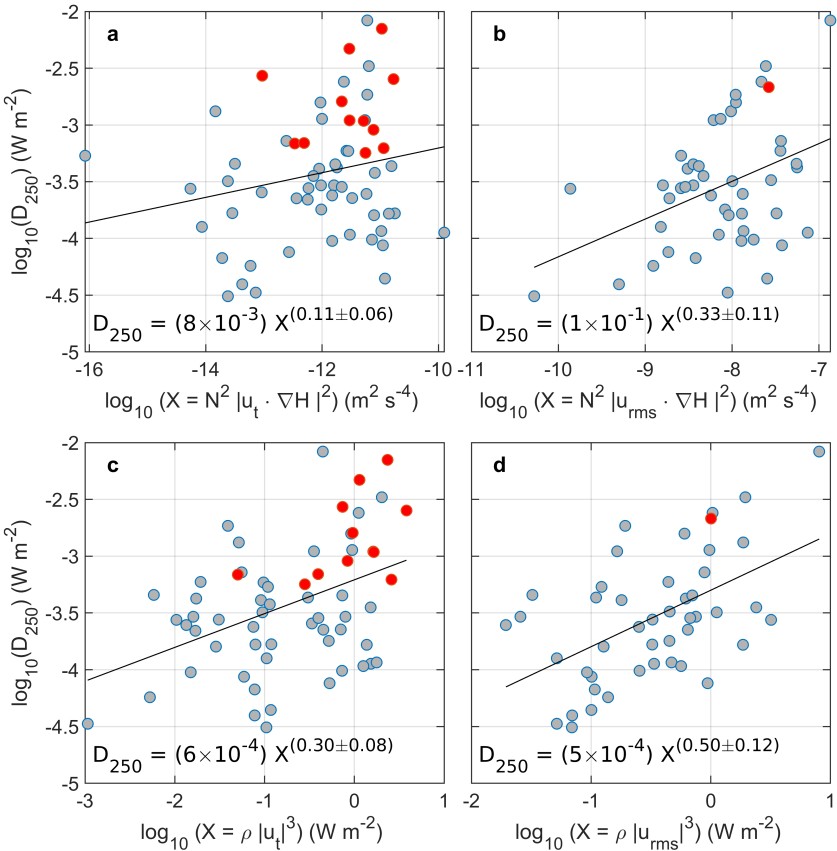

**Figure 9.** Depth-integrated dissipation rate in the bottom 250 m ($D_{250}$) regressed against a) the instantaneous (using $u_t$) values of a tidal-work related parameter, b) the statistically representative (using $u_{rms}$) typical values of the same parameter, c) the instantaneous tidally-driven bottom drag, and d) the typical tidally-driven bottom drag (using $u_{rms}$). See text for details. Linear fits on logarithmic parameter space (i.e., power-law fits) are the black lines and the corresponding equations are indicated with the 95% confidence levels. Red dots are the data points from the RS2 station. Process stations are batch-averaged (in sets of 4-5 consecutive profiles) in panels a and c, and averaged over the station duration in panels b and d.

## 7  Discussion

### 7.1  Pan-Arctic estimates of tidally-driven dissipation rates

Turbulent mixing in the Arctic is not well-documented, and measurements close to the bottom are scarce. The bottom drag
estimated from the Arc5km2018 predictions in the Arctic Ocean, using a constant drag coefficient, is larger on the shelf and on the ridges than in the deep basin (not shown), as a result of sensitivity to the strength of barotropic tidal current. These areas coincide with regions of enhanced tidal activity in the Arctic (Padman and Erofeeva, 2004). Using this tidally-driven bottom



drag and the relation inferred from the data collected north of Svalbard (section 5.3 and equation in Figure 9d), we estimate the depth-integrated dissipation rate. The highest bottom depth-integrated dissipation rates in the Arctic are found on the shelves,

and are consistent with the pan-Arctic observations compiled and presented in Rippeth et al. (2015), reaching $10^{-3}$ W m$^{-2}$ (not shown).

Because the parameterization is obtained using a limited data set from a localized region north of Svalbard, instead of presenting Arctic-wide maps we concentrate on the Eurasian Basin from north of Svalbard into the East Siberian Sea. The cross-isobath tidal currents along this transect, particularly in the Laptev Sea, are strong (see Fig. 1 of Fer et al. (2020b)).

Figure 10 shows the time-averaged cross-isobath tidal current amplitude and the 250 m bottom depth-integrated dissipation rate estimated using the equation in Figure 9d along the continental slope of the Eurasian basin. The largest tidal speeds are observed north of Svalbard and in the eastern part of the Laptev Sea where the slope connects to the Lomonosov Ridge, reaching more than 0.1 m s$^{-1}$ (Figure 10a). The largest average bottom dissipation rates across the continental slope are observed at 35°E, just east of Svalbard and at the Lomonosov Ridge, reaching $3.2 \times 10^{-4}$ W m$^{-2}$. We present two estimates for the

dissipation: vertically-integrated dissipation rate in the bottom 250 m, $D_{250}$, averaged laterally between the 400 m and 1200 m isobaths (blue line and left axis, Figure 10b), and $D_{250}$ integrated meridionally between the 400 m and 1200 m, isobaths (red line and right axis, Figure 10b). This volume-integrated dissipation rate, per unit metre along the shelf break, shows variations similar to the averaged $D_{250}$, except at 70°E. This is the location of the Santa Anna Trough, where the Atlantic Water from the Barents Sea flows into the Arctic Ocean and where the distance between the 400 and 1200 m isobaths triples compared

to the rest of the Eurasian continental slope. Rippeth et al. (2015) argued, based on microstructure measurements and tidal velocities from the TPXO8 inverse solution that the energy supporting much of the enhanced dissipation along the continental slopes in the Eurasian Basin, and more specifically north of Svalbard and around the Lomonosov Ridge, is of tidal origin. The mean-integrated dissipation over the Atlantic layer observed in Rippeth et al. (2015) is of similar order of magnitude as the depth-integrated dissipation in the bottom 250 m deduced from the tidally-driven bottom drag as observed in our study.

However, the bottom depth-integrated dissipation rate extrapolated from a local relation valid north of Svalbard to the Arctic Ocean must be considered with caution.

As the Eurasian Arctic is poleward of the critical latitude for most of the main tidal constituents, the response to tidal flow over sloping topography can be nonlinear when the topographic obstruction of the stratified flow is large. Legg and Klymak (2008) proposed that an inverse Froude number, $Fr_\omega^{-1}$, based on a vertical excursion distance of the tidal current over bottom

slope, can be used to estimate the possibility of occurrence of highly nonlinear jump-like lee waves, such as those observed at station RS2 (Fer et al., 2020b) or modelled over the Spitsbergen Bank (Rippeth et al., 2017), a shallow bank south of Svalbard and poleward of the M$_2$ critical latitude. The inverse Froude number is expressed as:

$$Fr_\omega^{-1} = \frac{|\nabla \mathbf{H}|\, N}{\omega} \tag{9}$$

where $|\nabla \mathbf{H}|$ is the bottom slope and $\omega$ is the tidal frequency. In our calculations we used the Brunt Väisälä frequency $N$

near the bottom, extracted from the MIMOC climatology in August (the result is not sensitive to this choice as the near

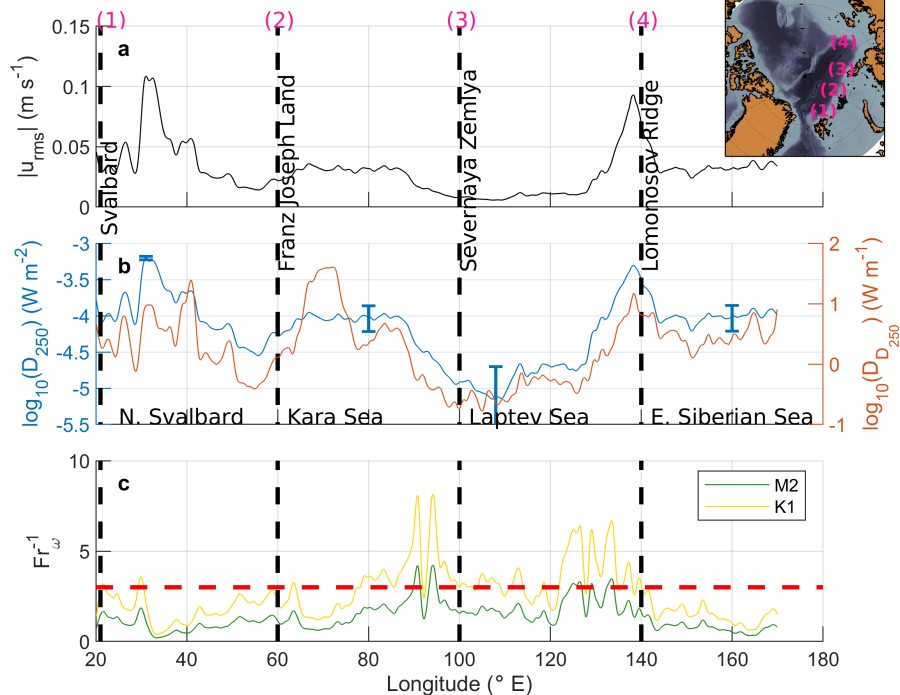

**Figure 10.** a) Typical cross-isobath tidal speed along the Eurasian continental slope obtained from Arc5km2018 and averaged meridionally between the 400 m and 1200 m isobaths b) left axis: the depth-integrated dissipation rate in the bottom 250 m, $D_{250}$, calculated from the tidally-driven bottom drag using the relation in Figure 9d, averaged between the 400 m and 1200 m isobaths. The blue vertical bars, shown at selected locations for clarity, are the error bars from the uncertainty on the equation in Figure 9d. Right axis: $D_{250}$ integrated laterally between the 400 m and 1200 m isobaths. c) Inverse Froude number for both the semi-diurnal $M_2$ (green) and the diurnal $K_1$ tidal components (yellow). The red dashed line is $Fr_\omega^{-1} = 3$, a threshold for the development of nonlinear processes (Legg and Klymak, 2008).

bottom stratification does not have a strong seasonal cycle). When $Fr_\omega^{-1} > 3$, the vertical excursion distance induced by tidal currents is sufficiently large that hydraulic jumps could occur and nonlinear waves can develop (Legg and Klymak, 2008). The calculations along the Eurasian shelf break are presented in figure 10c for the semidiurnal and the diurnal tidal forcing. Along the Eurasian slope, we expect nonlinear internal waves to develop for the diurnal tidal forcing in the eastern part of the Kara and

Laptev Seas, where $Fr_\omega^{-1}$ is much larger than 3, the threshold value for the development of nonlinear processes. Values slightly above this threshold are also seen north of Svalbard, consistent with the observations of nonlinear internal waves documented in Fer et al. (2020b) during the July cruise at RS2. They showed abrupt isopycnal vertical displacements of 10–50 m and an intense dissipation associated with cross-isobath diurnal tidal currents of $\sim 0.15$ m s$^{-1}$. The dissipation of these nonlinear internal waves creates an increase in dissipation in the whole water column by a factor of 100 and turbulent heat fluxes are

about 15 W m$^{-2}$ compared with the background turbulent heat flux of 1 W m$^{-2}$ (Fer et al., 2020b).





## 7.2 Impact of the melt water on the near-surface mixing

The observations used in this study indicate that although dissipation rates at the base of the mixed layer are similar from summer to fall, the downward buoyancy and upward salinity fluxes are larger in September due to an increase in the meltwater content.

In the future, sea ice meltwater is expected to increase and turbulent mixing near the surface to decrease. But at the same time, the increase in melt water is associated with a decrease in the sea ice cover, hence a possible increase in the momentum transfer from the wind into the ocean and a possible increase in turbulence (Dosser and Rainville, 2016). The balance between these two trends, increase in sea ice melt and increase in the open ocean area, with opposite effect on the turbulent mixing was tentatively addressed looking at a 1D model in Davis et al. (2016). They found that in the Eurasian Basin within a decade an
elevated shear will deepen the halocline and strengthen the stratification over the Atlantic Water thermocline, thus reducing the vertical heat flux. But in a long-term perspective, the mixed layer begins to cool and the sea ice cover can only be significantly affected if the elevated mixing is sufficient to erode the stratification barrier associated with the cold halocline. Using a steady-state vertical advection-diffusion balance and an exponential fit to the salinity and density profiles from an annual climatology, Fer (2009) estimated that a basin-averaged diffusivity larger than $5 \times 10^{-5}$ m$^2$ s$^{-1}$ would erode and eventually remove the
steady-state cold halocline layer.

Several studies also suggest that the meltwater will also have consequences on the nitrates fluxes (Randelhoff et al., 2017). Randelhoff et al. (2016) found that Atlantic Water heat leads to stronger melt rates north of Svalbard and an earlier onset of stratification which might be indirectly linked to bloom development.

## 7.3 Atlantic Water heat loss

The Atlantic Water looses heat as it propagates cyclonically along the continental slope in the Arctic Ocean. Around the Yermak Plateau, the along-path cooling and freshening are estimated to be $0.2°$C per 100 km and 0.01 g kg$^{-1}$ per 100 km, corresponding to a surface heat flux between 400 and 500 W m$^{-2}$ (Boyd and D'Asaro, 1994; Cokelet et al., 2008; Kolås and Fer, 2018). We found that the upward heat loss from turbulent heat fluxes from the Atlantic Water layer up to the mixed layer reached on average 8 W m$^{-2}$. This figure is one order of magnitude larger than vertical heat flux from the Atlantic Water to
the surface in the Laptev Sea (on the order of 0.1 - 1 W m$^{-2}$, Polyakov et al. (2019)). North of Svalbard and in the Laptev Sea, heat loss due to turbulent vertical mixing represents less than 10% of the total heat loss of the Atlantic Water (Kolås et al., submitted; Polyakov et al., 2019).

Vertical turbulent heat fluxes are not the main source of cooling of the Atlantic Water layer in the Arctic. Ivanov and Timokhov (2019) reviewed that from the Yermak Plateau to the Lomonosov ridge, 41% of the Atlantic Water heat is lost to the
atmosphere, 31% to the deep ocean and 20% is lost laterally.

North of Svalbard, a particular sink for the Atlantic Water heat content is eddy spreading from the slope into the basin (Crews et al., 2018; Våge et al., 2016). Using eddy-resolving regional model results, Crews et al. (2018) found that eddies export 1.0 TW out of the boundary current, delivering heat into the interior Arctic Ocean at an average rate of $\sim 15$ W m$^{-2}$.





Large heat losses during extreme events should not be ignored. For example, Meyer et al. (2017) found that at steady state,
heat fluxes across the 0°C isotherm are about 7 W m$^{-2}$; however, turbulent heat fluxes during storms can exceed 30 W m$^{-2}$.
West of Svalbard, Kolås and Fer (2018) found that the measured turbulent heat flux in the WSC is too small to account for its
cooling rate, but reported substantial contribution from energetic convective mixing of an unstable bottom boundary layer on
the slope. Convection was driven by Ekman advection of buoyant water across the slope, and complements the turbulent mixing
in the cooling process. The estimated lateral buoyancy flux is about $10^{-8}$ W kg$^{-1}$, sufficient to maintain a large fraction of the
observed dissipation rates, and corresponds to a heat flux of approximately 40 W m$^{-2}$. Although this process is documented
west of Svalbard, we can expect similar processes to occur north of Svalbard and extract heat and salt from the Atlantic Water
core.

## 8  Summary

We reported on observations of turbulence north of Svalbard, collected during two cruises in summer and fall 2018, in condi-
tions with varying tidal forcing and weak to moderate wind forcing with low variability. We describe the observed structure
of dissipation rates and vertical mixing in the region and identify the main processing supplying energy for turbulence. This
dataset complements the scarce observations and offers further insight to turbulent mixing processes in the Arctic Ocean.

During the observation period, from July to September, the surface meltwater content increases. Averaged across the base of
the mixed layer, salt and buoyancy fluxes more than double from summer to fall, although the vertically-integrated dissipation
rate in the mixed layer ($D_{ml}$) remains similar. Variability of the turbulent dissipation in the mixed layer varies nonlinearly with
the energy input from the wind $E_{10}$, approximated by $D_{ml} \propto E_{10}^{1.4}$. The scatter is large, however, from turbulence produced in
the mixed layer by other processes such as convection.

In the deeper part of the water column, tidal forcing appears to be one of the main sources of mixing. When the tidal current
amplitude exceeds 5 cm s$^{-1}$, near-bottom dissipation rates and diapycnal diffusivity double. The vertical decay scale of the
diffusivity is 22 m, compared to 18 m for weaker tidal currents. The variability of the vertically-integrated dissipation rate in
the bottommost 250 m, $D_{250}$, can be approximated by bottom stress from the barotropic tidal current, parameterized using a
quadratic bottom drag. Using the cross-isobath component of the tidal currents predicted over 30 days from Arc5km2018, re-
gression of $D_{250}$ against the tidally-driven bottom drag $W_{botdrag}$ gives $D_{250} \propto W_{botdrag}^{0.50}$. The average bottom drag coefficient
north of Svalbard is estimated to be about $8 \times 10^{-4}$. Applying the power-law fit to tidal currents along the Eurasian continental
slope, we find that turbulence is enhanced north of Svalbard and east of the Laptev Sea above the Lomonosov ridge, reaching
$3.4 \times 10^{-4}$ W m$^{-2}$. Higher above the seafloor, the dissipation rates can also increase as a result of breaking nonlinear internal
waves driven by tidal currents. A Froude number based calculation suggests that nonlinear response and internal hydraulic
jumps are expected to develop north of Svalbard, in the Kara and Laptev Seas. The generalisation of our results to the Eurasian
Basin should however be considered with caution as it is based on an empirical relation extrapolated from north of Svalbard.
More in situ observations from different sites are needed to confirm our results.



The Atlantic Water layer north of Svalbard cools and freshens by mixing with the surrounding waters. Heat loss due to vertical turbulent heat fluxes from the top of the Atlantic Water layer to the mixed layer is the largest, reaching $\sim 30\,\mathrm{W\,m^{-2}}$, above the 800 m isobath, corresponding to the location of the Atlantic Water boundary current core. In our dataset, the heat loss from the Atlantic Water layer due to vertical mixing is about $5\,\mathrm{W\,m^{-2}}$ and accounts for only about 1% of the total heat loss

of the Atlantic Water layer. Although the vertical turbulent heat fluxes are expected to increase during storms, more integrated studies addressing lateral mixing as well as vertical mixing are needed to close the heat budget in this region.

*Data availability.* All data are available from the Norwegian Marine Data Centre; data sets from the July cruise (KB 2018616) are available at https://doi.org/10.21335/NMDC-2047975397, data sets from the September cruise (KH 2018709) are available at https://doi.org/10.21335/NMDC-2039932526.

*Author contributions.* ZK drafted the manuscript. All the authors collected, processed and analyzed the observation, edited and commented on the manuscript.

*Competing interests.* Ilker Fer is a member of the editorial board of Ocean Science, but other than that the authors declare no competing interests.

*Acknowledgements.* This work was supported by the Nansen Legacy Project, project number 27272. We thank the officers, crew and scien-
tists of the *R/V Kronprins Haakon* cruise in September 2018 and of the *R/V Kristine Bonnevie* cruise in July 2018.





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
