# Peer review of "Structure and drivers of ocean mixing north of Svalbard in summer and fall 2018"

_Ocean Science, 2020_

## Referee Comment (RC1) · Anonymous Referee #1 · 22 Sep 2020

This manuscript presents a particularly interesting set of turbulence observations from north of Svalbard in the Arctic that cover the 2018 summer and autumn period. The authors investigate the vertical structure of mixing, heat fluxes and seasonal changes, and identify the processes driving the variability in the turbulence field. Both the wind and tidal supplies of energy are estimated with parameterizations derived and discussed. An attempt to extrapolate to the whole Eurasian Basin is made and interesting areas are identified that could be investigated in further work. The current lack of turbulence measurements in the Arctic is highlighted as the main limitation to pan-Arctic parameterization as well as the difficulty in accounting for lateral processes and fluxes, and for extreme events such as storms.

The quality of the English in the text is excellent. The Abstract and Introduction are

good, the Data & Methods and Observations sections are excellent. The Upper layer Dynamics section is fine. The section on Mixing in the AW layer is very interesting. The Tidal Mixing section presents a very nice analysis and tools. The Discussion is hard to link to this particular study's findings. The Summary section is excellent. The figures are excellent and have great detail. It was a pleasure to read through this work.

Major comments:

-Introduction: Sort out the introduction part on the various sources and intensity of turbulence in the Arctic (see individual comments further down).

-Discussion: Currently the discussion section reads in parts (see individual comments) more like a literature review than a discussion around how your findings fit in current research and their wider impact and implications. You have excellent results and just need to rewrite this section a little.

In its current form, the manuscript is already very good and presents a trove of findings for this region on the topic of turbulence. However, the manuscript would benefit from some sorting in parts, better highlight of key findings throughout (done well in the Summary), and better framing of this study's results in the discussion. I recommend that the manuscript is accepted subject to minor revision and look forward to seeing a revised version.

———————————————————————————————————————————— Individual comments

Abstract:

- Well written overall. The first sentence could do with rewriting to better reflect the beginning of your introduction. Right now you fit too much in that sentence and loose some of the meaning.

1.Introduction:

- L23: You state 'In the near future we may enter a new regime, in which the interior Arctic Ocean is entirely ice free in summer and sea ice is thinner and more mobile in winter'. I would argue that 'may' here is inappropriate and 'will' is more suitable. 'May' creates doubts around the likelihood of this happening. Please rephrase to better reflect current research findings such as the latest estimate from Guarino et al. (Guarino, M., Sime, L.C., Schröeder, D. et al. Sea-ice-free Arctic during the Last Interglacial supports fast future loss. Nat. Clim. Chang. (2020). https://doi.org/10.1038/s41558-020-0865-2) of 2035 for first ice free summer, or average from CMIP6 models of 2046 with a range of roughly 2030-2065.

- L31-37 and L38-46: In both these paragraphs, you describe the various sources and intensity of mixing in the Arctic. These two sections could do with merging and a better ordering of the different sources and intensity discussed.

- L65: Consider adding the following reference somewhere here: 'The lack of sea ice is mainly due to heat from the Atlantic layer reaching the surface'. Duarte, P., Sundfjord, A., Meyer, A., Hudson, S. R., Spreen, G., & Smedsrud, L. H. (2020). Warm Atlantic water explains observed sea ice melt rates north of Svalbard. Journal of Geophysical Research: Oceans, 125, e2019JC015662. https://doi.org/10.1029/2019JC015662

2. Data and Methods:

- L105: Unclear what 'In total, we collected 31 profiles.' Do you mean ship CTD profiles? Or VMP profiles or ? This doesn't match other number of VMP profiles stated earlier in the manuscript.

- L126: Pls define 'g' in equation (4) if not previously defined.L129-130: You state here that 'We used the profiles collected from the ship's CTD system (Sea-Bird Scientific, SBE 911plus on both cruises) to check and correct the temperature and salinity from the VMP'. But earlier on L107 you state 'A good agreement was observed and no correction was made.'. Please rewrite to make both statement consistent.

3.Overview of observations:

- L172-173: Unsure you need this statement here considering you have explained it clearly it in the figure caption. - Figure 3: Add what the red line is MLD in the caption.

4. Upper layer dynamics:

- L252: Add the definition of Dml in the text. Currently it only appears in Fig.6 caption. Can you make it clearer in the text how you obtained your estimate of the relationship between Dml and E10: it's a linear fit of Dml from the VMP data and E10 from the ship wind speed measurements.

5. Mixing in the Atlantic Water layer:

- L264: Should 'in present conditions of a warming Arctic' not be 'in the new conditions of a warming Arctic'?

- Fig.7 is great

- L274-275: This statement is confusing 'vertical turbulent heat fluxes are negative (less than 5Wm-2)' You might want to rephrase to 'vertical turbulent heat fluxes are negative (0 to -5Wm-2)'

- L282: Which section are you speaking about when you say '... the heat loss due to vertical turbulent heat fluxes is about ... across the section'?

- L282-285: Why is your estimate of heat loss due to vertical turbulent heat fluxes (1.2x10ˆ5 W/m) so much lower than Kolas estimates from the same cruise (9.1x10ˆ7 W/m and 9.6x10ˆ6 W/m)?

6. Tidal mixing:

- Fig. 8 caption: 'Average profiles of a) dissipation rate, b) turbulent heat flux and c) diapycnal diffusivity k for small' Also add Espi and F_H after the variable's names.

- L326-366: Nice analysis of the vertically integrated dissipation rate in bottom 250m.

[Figure]

7. Discussion:

- Fig.10 caption: I suggest removing the first word 'Typical'. Also, what is the background shading on the small map, topography? This map is useful and should be listed in the caption.

- L358: Subsection title 'Pan-Arctic estimates of tidally-driven dissipation rates' is not representative of results presented which are 'instead of presenting Arctic-wide maps we concentrate on the Eurasian Basin from north of Svalbard into the East Siberian Sea'. Please change section title to represent better the content. Also edit L355 in the previous section announcing the 'pan-Arctic estimate'.

- L398-405: Great findings.

- L410: Rephrase sentence 'In the future, sea ice meltwater is expected to increase and turbulent mixing near the surface to decrease' to better justify/explain the expected decrease in mixing (due to increase stratification).

- L 423: 'and an earlier onset of stratification which might be indirectly linked to bloom development' . . . due to . . . . Please add details.

- Section 7.2: I m unsure about the contribution your results make in this theme of 'impact of meltwater on the near surface mixing'. Consider better linking to your observations or moving this section as context in your introduction in a condensed form.

- L433: I m unsure about how this statement 'Vertical turbulent heat fluxes are not the main source of cooling of the Atlantic Water layer in the Arctic. Ivanov and Timokhov (2019) reviewed that from the Yermak Plateau to the Lomonosov ridge, 41% of the Atlantic Water heat is lost to the atmosphere, 31% to the deep ocean and 20% is lost laterally.' fits with the previous 'heat loss due to turbulent vertical mixing represents less than 10% of the total heat loss of the Atlantic Water' . Would the 10% not be part of the 31% deep ocean and 20% laterally? You seem to imply they are different when you state 'Vertical turbulent heat fluxes are not the main source of cooling of the

Atlantic Water layer in the Arctic'. Please tidy up these two paragraphs so the reader can follow your thoughts. Again, further down you discuss eddies and their roles. But is the heat export from eddies not included in the 20% lost laterally from Ivanov and Timokhov (2019)?

- L444 and 445: The numbers you quote there ($10^{-8}$ and $40W/m^2$), are they from Kolas and Fer or from this study? Again, how does this section of the discussion (7.3 AW heat loss) exactly links with your findings. Currently this reads a lot like an (excellent) literature review, rather than you putting your new findings in context. . .

8. Summary:

- L459-460: Consider adding 'The vertical decay scale of the diffusivity is 22m *for those strong tidal currents*, compared to 18m for weaker tidal currents.'

- L470: Consider adding details 'More in situ observations from different sites *in the Eurasian Basin and elsewhere in the Arctic* are needed to confirm our results.'

- L475: Can you add 'of the *expected/estimated* total heat loss of the Atlantic Water layer'.

- L475-476: Can you explain better the relation between the first part of the sentence and the later part? I understand you mean to say that increased vertical mixing during storms might partially close the budget but don't make up the whole 'missing' heat loss which might be mostly lateral fluxes. So that both lateral fluxes and extreme conditions such as storms, frontal systems etc should be investigated. But this will not super clear in the current form of the sentence.

---

## Short Comment (SC1) · 5 Oct 2020

We will submit a detailed point-by-point response to all reviewers' comments when the discussion closes. The first author, Zoe Koenig, is presently on the MOSAiC expedition with limited internet access. Ilker Fer coordinated with Zoe to get the first response to reviewer 1's comments, for the sake of discussion.

Thank you very much for your very helpful and thoughtful comments. We will take them into consideration when preparing the revised version. We will improve the text following your comments, checking that the numbers match and are consistent throughout the text, adding clarification where needed, and considering the new references suggested by the reviewer.

[Figure]

In the revised version, we will restructure the introduction and the discussion as suggested. We will rearrange the introduction paragraph that describes the different sources of mixing in the Arctic Ocean. We will merge both paragraphs (l.31-37 and 38-46) and we will first summarize the different sources of mixing and then go through their intensity.

In the discussion, we will delete subsection 7.2 'Impact of the melt water on the near-surface mixing', as we agree that our results do not contribute much to that section.

We will change the title of subsection 7.1 to 'Estimates of tidally-driven dissipation rate in the Eurasian Basin' as we do not show pan-Arctic estimates.

Regarding section 7.3 'Atlantic Water heat loss', we agree that it looks more like a literature review and that our results are not linked appropriately to the actual discussion. We will rearrange this subsection completely. We agree that the sentence l.433 is ambiguous. We found that turbulent vertical mixing represents less than 10% of the total heat loss of the Atlantic Water layer, but indeed we do not specify where the heat is lost, so these 10% are not to be compared with the percentages from Ivanov and Timokhov (2019). We will clarify this notion. Although the subsection resembles a literature review, we intend to retain it, but we will link it better to our findings.

---

## Referee Comment (RC2) · Anonymous Referee #2 · 7 Oct 2020

General comments

The authors present a series of observations of turbulent dissipation from measurements taken during two separate research cruises in the region along the slope north of Svalbard. The study considers wind forcing and tides as drivers to mix heat that is concentrated in warm Atlantic-origin water that resides in the mid-depths of the water column. Vertical profiles of turbulent dissipation, diffusivity, and heat and buoyancy fluxes are presented and tied to seasonal changes and input work from both winds and tides. Near the end of the paper, the authors extrapolate their ideas across a broader region.

This manuscript makes an important addition to the body of literature on turbulence and mixing in a key Arctic region. While the results and analysis are interesting and

merit publication, the manuscript would be greatly improved by more cohesive linking of the different ideas. As presented, the study reads as a nice collection of related results, but parts of the discussion do more to highlight some of the background and motivation than to link to those results, and many of the results are considered independently despite parallels in the analysis. Consequently, the study lacks a coherent story. There is enough detail in the manuscript already that this should not require any further analysis, but the authors should consider some reorganization of the discussion section to tie together different aspects of the study.

One potential approach to this reorganization would be to rethink the presentation of sections 4-6. Currently, these sections are organized to step vertically down through the water column from the upper ocean (§4) to Atlantic Water (§5), to the bottom boundary layer (§6). However, the wind forcing and tides are presented as the main drivers of vertical mixing while, in some capacity, the Atlantic Water is what is being mixed. It may be better to move some of the ideas from section 5 to the discussion, and use it to unify and compare/contrast the different results from sections 4 and 6 (e.g., is the structure seen in figure 7 a consequence of the results in sections 4 or 6?). Then presenting wind forcing and tidal forcing back-to-back will better highlight the parallels between the analysis in each of those sections.

Specific comments

- L106-107: Here you state that VMP measurements of temperature and salinity agreed with ship CTD profiles, so no corrections were made. But in L129-130 you discuss using the ship CTD to correct the VMP temperature. Please ensure your statements are consistent.

- L116-122: There are a wide number of parameterizations and methods for determining diapycnal diffusivity. You should discuss the sensitivity of your results to the choice of the Bouffard and Boegman (2013) method compared (at least) to the more common Osborn (1980) method with Gamma = 0.2.

- L134: The measurement height for wind speed should be mentioned here, along with the correction to 10m, instead of L248-249.

- L135-137: Have tidal current measurements from the Arc5km2018 model been verified in this region?

- L143-146: These lines about the number of profiles could be moved into the methods section (2.1).

- L156-160: These lines seem out of place here.

- L178-180: Equation 5 doesn't represent the surface layer depth from Randelhoff et al. (2017). It represents the scaled vertical coordinate those authors use, and the surface layer depth corresponds to a specific value of r. This isn't clear in your text. - L178-183: There is a lot of detail here for a surface layer definition that you ultimately don't use. This could be simplified by trimming out a number of intervening sentences and leaving only the beginning and end: "We also estimate a surface layer depth following Randelhoff et al. (2017); however, the mixed layer depth and the surface layer depth are very similar (not shown), so in the rest of the study...".

- L186-187: This warm water is difficult to identify in the sections (especially panels a and c).

- L191-193: Are these averages of profiles from both July and September cruises? If not, which set are these? Please clarify in your text. Also, if surface stratification and buoyancy flux are significantly different in July and September (e.g., section 4.1), then I would expect the shallow parts of those profiles to be fairly distinct between seasons and not appropriate to average. Ensure that you comment on that in the text.

- Figure 3: The red line showing mixed-layer depth is very difficult to see. In the left panels, it blends into the temperature field and in the right panels it is obscured by other details. This is also partly due to how close to the surface the mixed layer is relative to the scale of the plot. In the left panels, the scale in the upper 100m differs from the

rest of the plot to better show upper ocean details, but they are still hard to see and the scale change could be further exaggerated. I did not immediately realize that the vertical scale change was not included in the right panels. I would also appreciate if the Atlantic water was somehow better identified or more visible in this figure – I don't clearly see it in all sections.

- Figure 4: Some of the subfigure tick-labels are overlapping and hard to read. The legend is small and difficult to read. Also, it may be helpful to replace legend labels with "inshore, shelf break, offshore" as are used in the text. It's difficult to see the details in the upper water column (below ∼100m); you may consider using a different vertical scale (as in figure 3), or providing insets that zoom in on the surface of each panel.

- L206-208: Here you say that the core of the Atlantic water current is between 400 m and 600 m, but in L186-187 you associate the Atlantic water with 500 m to 1100 m depths. Throughout the text you use the 800 m isobath as a reference for Altantic water, which is consistent with L186-187 but not with L206-208. Please clarify this and ensure consistency throughout.

- L208: The only mention of current measurements throughout the rest of the paper are the modeled tidal currents, but this sentence is about water column currents. Are these measured with a shipboard ADCP during the cruise? Or is this sentence a reference to known characteristics of the Atlantic water layer from other studies (e.g., the submitted work by Kolås et al., that you reference in L196)?

- L250-256: The non-linear dependence of mixed-layer dissipation on wind energy input is a really interesting result, but it would be valuable to explore this concept in more detail and relate it to prior studies (either here, or in some part of the discussion). In particular, there has been some theory that looks at this relation in the wave-boundary layer and may or may not support a linear relationship (see Craig & Banner, 1994, doi: 10.1175/1520-0485(1994)024<2546:MWETIT>2.0.CO;2 and Thomson, 2016, doi:

10.1175/JPO-D-15-0130.1). It's also been considered in a bulk sense in the mixed layer (i.e., as an efficiency; see Sutherland et al., 2013 doi:10.5194/os-9-597-2013 and references therein; though this is still in the wave-breaking framework). I think there might be some richness in the fact that this analysis suggests a non-linear relationship and worth speculating about why or what that might mean (perhaps stratification or mixed-layer depth play in in some way). Additionally, it may be worth mentioning the wave conditions during the sampling in section 3.1, even if only qualitatively.

- Section 4.2: In this section you take all of the data together, but in section 4.1 you contrast some of the details of the mixed-layer between the July and September cruises. Am I correct in interpreting from figure 6 and L252-253 that there's not enough data in September to be able to be able to make meaningful comparisons of Dml between seasons? If it's possible to contrast the effects of wind forcing between the two seasons at all, it would be very interesting.

- Figure 6: Are there any noticeable relationships if you colour the points by mixed-layer depth?

- L316: What are the confidence intervals on the decay scales? Are 18 m and 22 m statistically different from each other?

- L320: "We investigate the role of two distinct contributions from tidal currents". Contributions to what? This sentence isn't clear.

- L326: Why the choice of 250 m for integrating the dissipation? Is this choice informed in any way by the estimated decay scales from earlier? Are results sensitive to other choices?

- L330-331: Why exclude wave (tidal) frequency?

- L327-347: Since the tidal-work parameter in equation 7 doesn't provide a useful correlation, you could choose to simplify the text and figure 9 by removing some of the text in the section, and simply stating that you also tried comparing D250 with the rate
of work given by Nash et al., (2006) but found no significant correlation. Then you could remove equation 7 and some of the text surrounding it and remove panels a and b from figure 9. This is a personal choice, but it would better highlight your positive results.

- L335: It's worth highlighting somewhere that equation 8 is analogous to the equation for E10 (in section 4.2), and so the nonlinear relationship between D250 and Wbotdrag is something that can be related to the nonlinear relationship between Dml and E10.

- L350: Does it make sense to compare the bottom drag coefficient to one from the Bering Strait? I wouldn't assume that the bottom morphologies of the two areas would be similar. Can you instead refer to a range of "typical" bottom drag values?

- Figure 9: I don't quite understand why there is only one red dot in panels b and d but many in panels a and c? Is this due to how you perform the u_rms calculation?

- L367-368: If you are showing only a line along the Eurasian Basin, then "Pan-Arctic" in the section title is not appropriate. (Note, the authors have already expressed plans to rename this section).

- L399-400: Maybe highlight to what extent the dissipation in panel b of Figure 10 will account for the nonlinear waves (e.g., if it did account for it, I'd expect to see peaks in D250 in panel b the correspond to the peaks of inverse Fr in panel c). Make it explicit that these are areas that warrant specific further study.

- L387-405: While the discussion of these potential non-linear wave "hotspots" is very interesting, it feels somewhat disconnected from the rest of the study. Most of the times non-linear waves present in the results before this point are references to the high dissipation event at RS2 that was already documented by Fer et al. (2020b). This section would connect more if you make more explicit comparisons between the general results and the non-linear wave results (e.g., you have RS2 points in red in figure 9, which show the associated increase in D250, but those points are presented as more of a sidenote in the text L351-353 when there's potential to make more direct

comparisons).

- Section 7.1: Overall, this is a nice extension of the ideas in section 6.

- Section 7.2: As written, there is no clear link between the ideas in the section and the results you've presented. Do your results agree with or refute any of the studies you cite? Can they be compared at all? This section provides interesting background and motivation, but without linking it explicitly to your results it is not really a discussion section. (Note, the authors have already expressed plans to remove this section).

- Section 7.3: Similar to section 7.2, this provides good background but isn't otherwise well linked to the rest of the study.

- L459-460: The different values of kappa_bot may be a stronger result to highlight in the summary than the different decay scales (or maybe include both?)

Technical corrections

- L25-26: Awkward grammar/sentence structure in the sentence starting with "The heat reservoir. . .".

- L41-42: Do you have the correct reference for the sentence "Wind-driven momentum input. . ."? Is this meant to reference Rainville and Woodgate (2009) instead of Rainville and Windsor (2008)?

- L154: "encounter" should be "encountered"

- L317: In the sentence "We use kappa_bot. . .", should that instead be kappa_bg?

- L414: Awkward grammar/sentence structure in the sentence starting with "They found. . .".

---

## Author Comment (AC1) · 20 Nov 2020

This manuscript presents a particularly interesting set of turbulence observations from north of Svalbard in the Arctic that cover the 2018 summer and autumn period. The authors investigate the vertical structure of mixing, heat fluxes and seasonal changes, and identify the processes driving the variability in the turbulence field. Both the wind and tidal supplies of energy are estimated with parameterizations derived and discussed. An attempt to extrapolate to the whole Eurasian Basin is made and interesting areas are identified that could be investigated in further work. The current lack of turbulence measurements in the Arctic is highlighted as the main limitation to pan-Arctic parameterization as well as the difficulty in accounting for lateral processes and fluxes, and for extreme events such as storms. The quality of the English in the text is excellent. The

[Figure]

Abstract and Introduction are good, the Data & Methods and Observations sections are excellent. The Upper layer Dynamics section is fine. The section on Mixing in the AW layer is very interesting. The Tidal Mixing section presents a very nice analysis and tools. The Discussion is hard to link to this particular study's findings. The Summary section is excellent. The figures are excellent and have great detail. It was a pleasure to read through this work.

Thanks for these comments

Major comments: -Introduction: Sort out the introduction part on the various sources and intensity of turbulence in the Arctic (see individual comments further down).

Agreed. We rearranged the introduction as suggested below

-Discussion: Currently the discussion section reads in parts (see individual comments) more like a literature review than a discussion around how your findings fit in current research and their wider impact and implications. You have excellent results and just need to rewrite this section a little. In its current form, the manuscript is already very good and presents a trove of findings for this region on the topic of turbulence. However, the manuscript would benefit from some sorting in parts, better highlight of key findings throughout (done well in the Summary), and better framing of this study's results in the discussion. I recommend that the manuscript is accepted subject to minor revision and look forward to seeing a revised version.

We have rearranged the discussion as suggested below

Individual comments Abstract: - Well written overall. The first sentence could do with rewriting to better reflect the beginning of your introduction. Right now you fit too much in that sentence and loose some of the meaning.

We changed the first sentence of the introduction: 'The Arctic Ocean has major implications on global scale as the Arctic Ocean is a main sink for heat and salt. Ocean mixing contribute to this sink by mixing the Atlantic and Pacific-origin waters with surrounding

waters.'

1.Introduction: - L23: You state 'In the near future we may enter a new regime, in which the interior Arctic Ocean is entirely ice free in summer and sea ice is thinner and more mobile in winter'. I would argue that 'may' here is inappropriate and 'will' is more suitable. 'May'creates doubts around the likelihood of this happening. Please rephrase to better reflect current research findings such as the latest estimate from Guarino et al. (Guarino,M., Sime, L.C., Schröeder, D. et al. Sea-ice-free Arctic during the Last Interglacialsupports fast future loss. Nat. Clim. Chang. (2020). https://doi.org/10.1038/s41558-020-0865-2) of 2035 for first ice free summer, or average from CMIP6 models of 2046 with a range of roughly 2030-2065.

We changed 'may' to 'will' and we added the reference Guarino et al., 2020

- L31-37 and L38-46: In both these paragraphs, you describe the various sources and intensity of mixing in the Arctic. These two sections could do with merging and a better ordering of the different sources and intensity discussed.

We merged the two paragraphs and we ordered better the sources and intensities. We also removed part of the description of the sources and intensities as we found that it did not serve the rest of the manuscript.

- L65: Consider adding the following reference somewhere here: 'The lack of sea ice is mainly due to heat from the Atlantic layer reaching the surface'. Duarte, P., Sundfjord A., Meyer, A., Hudson, S. R., Spreen, G., & Smedsrud, L. H. (2020). Warm Atlantic water explains observed sea ice melt rates north of Svalbard. Journal of Geophysical Research: Oceans, 125, e2019JC015662. https://doi.org/10.1029/2019JC0156622.

We added the reference

2.Data and Methods: - L105: Unclear what 'In total, we collected 31 profiles.' Do you mean ship CTD profiles? Or VMP profiles or ? This doesn't match other number of VMP profiles stated earlier in the manuscript.

Thanks for spotting this mistake. We deleted this sentence

- L126: Pls define 'g' in equation (4) if not previously defined.

We added the definition of g: 'where alpha and beta are respectively the thermal expansion and salinity contraction coefficients, and g is the gravitional constant.'

L129-130: You state here that 'We used the profiles collected from the ship's CTD system (Sea-Bird Scientific,SBE 911plus on both cruises) to check and correct the temperature and salinity from the VMP'. But earlier on L107 you state 'A good agreement was observed and no correction was made.'. Please rewrite to make both statement consistent.

Thanks for pointing it out. We deleted 'correct' in the first sentence.

3.Overview of observations: - L172-173: Unsure you need this statement here considering you have explained it clearly it in the figure caption.

Agreed. We deleted this sentence

- Figure 3: Add what the red line is MLD in the caption.

We added in the caption that the (now) green line is the mixed layer depth.

4. Upper layer dynamics: - L252: Add the definition of Dml in the text. Currently it only appears in Fig.6 caption. Can you make it clearer in the text how you obtained your estimate of the relationship between Dml and E10: it's a linear fit of Dml from the VMP data and E10 from the shipwind speed measurements.

We added the definition of Dml and clarify that we apply a linear fit

5. Mixing in the Atlantic Water layer: - L264: Should 'in present conditions of a warming Arctic' not be 'in the new conditions of a warming Arctic'?

Changed as suggested.

- Fig.7 is great

Thanks!

- L274-275: This statement is confusing 'vertical turbulent heat fluxes are negative(less than 5Wm-2)' You might want to rephrase to 'vertical turbulent heat fluxes are negative (0 to -5Wm-2)'

Changed as suggested.

- L282: Which section are you speaking about when you say '...the heat loss due to vertical turbulent heat fluxes is about... across the section'?

We are talking about the cross-isobath section. We agree that 'across the section' is more confusing than helpful and we deleted it.

- L282-285: Why is your estimate of heat loss due to vertical turbulent heat fluxes(1.2x10ËĘ5 W/m) so much lower than Kolas estimates from the same cruise (9.1x10ËĘ7W/m and 9.6x10ËĘ6 W/m)?

Here we estimate the heat loss only due to vertical turbulent heat fluxes. Kolås et al. (2020) estimate the along-path change of heat content, that takes into account not only the vertical turbulent heat fluxes but also the other fluxes that can impact the heat content.

6. Tidal mixing: - Fig. 8 caption: 'Average profiles of a) dissipation rate, b) turbulent heat flux and c) diapycnal diffusivity k for small' Also add Espi and F_H after the variable's names.

Done

- L326-366: Nice analysis of the vertically integrated dissipation rate in bottom 250m.

Thanks

7. Discussion: - Fig.10 caption: I suggest removing the first word 'Typical'. Also, what is the back-ground shading on the small map, topography? This map is useful and

should be listed in the caption.

'Typical' reinforce the idea that we use u_rms. The background shading on the small map is topography, we added this information in the caption.

- L358: Subsection title 'Pan-Arctic estimates of tidally-driven dissipation rates' is not representative of results presented which are 'instead of presenting Arctic-wide maps we concentrate on the Eurasian Basin from north of Svalbard into the East Siberian Sea'. Please change section title to represent better the content. Also edit L355 in the previous section announcing the 'pan-Arctic estimate'.

We changed the title of the subsection to 'Estimates of tidally-driven dissipation rate in the Eurasian Basin' and we edited l.355: 'An Eurasian-basin coarse estimate will be given ...'.

- L398-405: Great findings.

Thanks

- L410: Rephrase sentence 'In the future, sea ice meltwater is expected to increase and turbulent mixing near the surface to decrease' to better justify/explain the expected decrease in mixing (due to increase stratification). - L 423: 'and an earlier onset of stratification which might be indirectly linked to bloom development'...due to.... Please add details. - Section 7.2: I m unsure about the contribution your results make in this theme of 'impact of meltwater on the near surface mixing'. Consider better linking to your observations or moving this section as context in your introduction in a condensed form.

We agree that this discussion is not really relevant to our analysis. We deleted section 7.2.

- L433: I m unsure about how this statement 'Vertical turbulent heat fluxes are not the main source of cooling of the Atlantic Water layer in the Arctic. Ivanov and Timokhov (2019) reviewed that from the Yermak Plateau to the Lomonosov ridge, 41% of the
Atlantic Water heat is lost to the atmosphere, 31% to the deep ocean and 20% is lost laterally.' fits with the previous 'heat loss due to turbulent vertical mixing represents less than 10% of the total heat loss of the Atlantic Water' . Would the 10% not be part of the 31% deep ocean and 20% laterally? You seem to imply they are different when you state 'Vertical turbulent heat fluxes are not the main source of cooling of the Atlantic Water layer in the Arctic'. Please tidy up these two paragraphs so the reader can follow your thoughts. Again, further down you discuss eddies and their roles. But is the heat export from eddies not included in the 20% lost laterally from Ivanov and Timokhov (2019)?

Yes, you are right. We are mixing different informations. We found that turbulent vertical mixing represents less than 10% of the total heat loss of the Atlantic Water layer, but indeed we do not specify where the heat is lost, so these 10% are not to be compared with the percentages from Ivanov and Timokhov (2019). We changed the sentence: ' Ivanov and Timokhov (2019) estimated that from the Yermak Plateau to the Lomonosov Ridge, 41% of the Atlantic Water heat is lost to atmosphere, 31% to deep ocean and 20% is lost laterally. Heat loss resulting from vertical heat fluxes contributes to the heat loss to atmosphere and to deep ocean, but not to the lateral heat loss. '

- L444 and 445: The numbers you quote there (10Ë̈Ę-8 and 40W/mË̈Ę2), are they from Kolas and Fer or from this study? Again, how does this section of the discussion(7.3 AW heat loss) exactly links with your findings. Currently this reads a lot like an (excellent) literature review, rather than you putting your new findings in context...

These numbers were from Kolås and Fer. We agree that this section looks more like a literature review, and we tried to better put our new findings in context. We mainly changed the last 2 paragraphs:

'West of Svalbard, Kolås and Fer (2018) found that the measured turbulent heat flux in the WSC was too small to account for the cooling rate of the Atlantic Water layer, but reported substantial contribution from energetic convective mixing of an unstable
bottom boundary layer on the slope. Convection was driven by Ekman advection of buoyant water across the slope, and complements the turbulent mixing in the cooling process. The estimated lateral buoyancy flux was about 10−8 W kg−1 (Kolås and Fer, 2018), sufficient to maintain a large fraction of the observed dissipation rates, and corresponds to a heat flux of approximately 40 W m−2. We can expect similar processes to extract heat and salt from the Atlantic Water core north of Svalbard. Such processes can explain why turbulent heat fluxes are only responsible for 10% of the Atlantic heat loss north of Svalbard. Furthermore, large heat loss during extreme events should not be ignored. For example, Meyer et al. (2017) found that the average heat flux of about 7 W m−2 across the 0âŮȩC isotherm increased during storms, exceeding 30 W m−2. During our survey without extreme wind events, the turbulent heat fluxes represent only a small portion of the heat loss of the Atlantic Water.'

8. Summary: - L459-460: Consider adding 'The vertical decay scale of the diffusivity is 22m *for those strong tidal currents*, compared to 18m for weaker tidal currents.'

Thanks, done

- L470: Consider adding details 'More in situ observations from different sites *in the Eurasian Basin and elsewhere in the Arctic* are needed to confirm our results.'

Thanks, done

- L475: Can you add 'of the *expected/estimated* total heat loss of the Atlantic Water layer'.

We added 'estimated'

- L475-476: Can you explain better the relation between the first part of the sentence and the later part? I understand you mean to say that increased vertical mixing during storms might partially close the budget but don't make up the whole 'missing' heat loss which might be mostly lateral fluxes. So that both lateral fluxes and extreme conditions such as storms, frontal systems etc should be investigated. But this will not super clear

in the current form of the sentence.

We reformulate the last sentence: 'Increased vertical mixing during storms would add to this figure. However, integrated studies addressing lateral mixing processes, frontal systems as well as extreme conditions such as storms are needed to close the heat budget in this region.'

Please also note the supplement to this comment:
https://os.copernicus.org/preprints/os-2020-77/os-2020-77-AC1-supplement.pdf

**Supplement:**

We thank the reviewers for their thoughtful comments and suggestions. We have addressed all comments. Below is our point-by-point response to the comments of both reviewers, reproduced in black, followed by our response in red.

**Response to referee #2**

General comments
The authors present a series of observations of turbulent dissipation from measurements taken during two separate research cruises in the region along the slope north of Svalbard. The study considers wind forcing and tides as drivers to mix heat that is concentrated in warm Atlantic-origin water that resides in the mid-depths of the water column. Vertical profiles of turbulent dissipation, diffusivity, and heat and buoyancy fluxes are presented and tied to seasonal changes and input work from both winds and tides. Near the end of the paper, the authors extrapolate their ideas across a broader region. This manuscript makes an important addition to the body of literature on turbulence and mixing in a key Arctic region.

Thanks for these comments! We are happy to read that our study is well-received.

While the results and analysis are interesting and merit publication, the manuscript would be greatly improved by more cohesive linking of the different ideas. As presented, the study reads as a nice collection of related results, but parts of the discussion do more to highlight some of the background and motivation than to link to those results, and many of the results are considered independently despite parallels in the analysis. Consequently, the study lacks a coherent story. There is enough detail in the manuscript already that this should not require any further analysis, but the authors should consider some reorganization of the discussion section to tie together different aspects of the study.

We agree that some reorganization was needed in the discussion to better highlight our results. The changes we made are described below in the point-by-point response to the reviewer. To summarize the main changes: we reordered the result sections (upper layer, tidal forcing and Atlantic water heat loss) for more coherence. We also reorganized the discussion as suggested by both reviewers.

One potential approach to this reorganization would be to rethink the presentation of sections 4-6. Currently, these sections are organized to step vertically down through the water column from the upper ocean (§4) to Atlantic Water (§5), to the bottom boundary layer (§6). However, the wind forcing and tides are presented as the main drivers of vertical mixing while, in some capacity, the Atlantic Water is what is being mixed. It may be better to move some of the ideas from section 5 to the discussion, and use it to unify and compare/contrast the different results from sections 4 and 6 (e.g., is the structure seen in figure 7 a consequence of the results in sections 4 or 6?). Then presenting wind forcing and tidal forcing back-to-back will better highlight the parallels between the analysis in each of those sections.

Thanks for this suggestion. We agree that having both forcing (wind and tide) sections following each other is a better structure, and revised accordingly. We decided to keep the section on the Atlantic Water heat loss as it is, and did not integrate it to the discussion. The material in this part is "results" and is not suitable to introduce in "discussion". We cannot attribute the structure seen in figure 7 to a consequence of the results in sections 4 and 6.

Specific comments

- L106-107: Here you state that VMP measurements of temperature and salinity agreed with ship CTD profiles, so no corrections were made. But in L129-130 you discuss using the ship CTD to correct the VMP temperature. Please ensure your statements are consistent.

Thanks for pointing this out. No correction was needed, we deleted 'correct' l. 130.

- L116-122: There are a wide number of parameterizations and methods for determining diapycnal diffusivity. You should discuss the sensitivity of your results to the choice of the Bouffard and Boegman (2013) method compared (at least) to the more common Osborn (1980) method with Gamma = 0.2.

The Bouffard and Boegman (2013) method differs from the Osborn (1980) method only for very low and very large Reynolds numbers. In our dataset, 80% of the Reynolds number falls between 8.5 and 400, and for this range of Reynolds number the diapycnal diffusivity is identical in both Bouffard and Boegman (2013) and Osborn (1980). Using Bouffard and Boegman (2013) resulted in fewer outliers, which is why this method was used. Our results are not sensitive to the choice of Bouffard and Boegman (2013) compared to Osborn (1980). We added in the text (after the introduction of Reb):
'In the transitional range (8.5<Reb< 400), calculation of \kappa is identical to Osborn (1980), using the canonical mixing coefficient of 0.2 (Gregg et al. 2018); however in the energetic regime the latter is an overestimate. In our dataset, 80% of the estimates are in the transitional regime.'

- L134: The measurement height for wind speed should be mentioned here, along with the correction to 10m, instead of L248-249.

Agree, we move this explanation l.134.

- L135-137: Have tidal current measurements from the Arc5km2018 model been verified in this region?

The Arc5km2018 has not been verified in this region. However, as far as we know, AOTIM5 (5km horizontal resolution Arctic Ocean Tidal Inverse Model) and its recent version developed in 2018 (Arc5km2018) are the best available estimates of the tidal forcing in the Arctic Ocean. Arc5km2018 has been improved compared to AOTIM5 as:
(1) it uses an improved prior model with ocean open boundary forcing from an updated TOPEX/Poseidon global barotropic global tide solution (TPXO9.1)
(2) it adds four tidal constituents, 2N2 and the three non-linear constituents
(3) it assimilates much longer time series of altimetry, notably from the ESA satellites that sample to 81.5 degrees north. (from https://arcticdata.io/catalog/view/doi:10.18739/A21R6N14K)

- L143-146: These lines about the number of profiles could be moved into the methods section (2.1).

Agreed, we moved these lines at the end of section 2.1.

- L156-160: These lines seem out of place here.

We agree that these lines are out of place here. We added these sentences rather to the introduction

- L178-180: Equation 5 doesn't represent the surface layer depth from Randelhoff et al. (2017). It represents the scaled vertical coordinate those authors use, and the surface layer depth corresponds to a specific value of r. This isn't clear in your text.

Yes indeed you are right, r is not the surface layer depth but rather a scaled depth coordinate, but as suggested in the following comment, we deleted these details.

- L178-183: There is a lot of detail here for a surface layer definition that you ultimately don't use. This could be simplified by trimming out a number of intervening sentences and leaving only the beginning and end: "We also estimate a surface layer depth following Randelhoff et al. (2017); however, the mixed layer depth and the surface layer depth are very similar (not shown), so in the rest of the study. . .".

Agreed. We deleted the details about the surface layer definition.

- L186-187: This warm water is difficult to identify in the sections (especially panels a and c).

We agree that it is a bit hard to identify the warm Atlantic Water in the panels. We now also refer to figure 4a (blue line) that show better the warm Atlantic Water. We also added a thicker 2 degree temperature contour in Figure 3 to emphasize the Atlantic Water layer.

- L191-193: Are these averages of profiles from both July and September cruises? If not, which set are these? Please clarify in your text. Also, if surface stratification and buoyancy flux are significantly different in July and September (e.g., section 4.1), then I would expect the shallow parts of those profiles to be fairly distinct between seasons and not appropriate to average. Ensure that you comment on that in the text.

The average is calculated using profiles from both July and September cruise. Surface stratification and buoyancy flux are indeed different in July and September, but these profiles are mainly used to discuss the deeper water column and not the upper 50 m. We now clarify this in the text.

'We calculated average profiles of temperature, salinity, dissipation rate and diffusivity using data combined from both July and September cruises. The averaging is made in isopycnal coordinates to account for the possible vertical displacement of isopycnals and water masses from the slope to the deep basin. Once averaged, the profiles are mapped onto vertical coordinate using the corresponding average depth of an isopycnal (Figure 4). While this averaging is representative of the vertical structure below the mixed layer, it is probably not appropriate for the surface layer where surface stratification and buoyancy flux are significantly different in July and September (see following section for more details).'

- Figure 3: The red line showing mixed-layer depth is very difficult to see. In the left panels, it blends into the temperature field and in the right panels it is obscured by other details. This is also partly due to how close to the surface the mixed layer is relative to the scale of the plot. In the left panels, the scale in the upper 100m differs from the rest of the plot to better show upper ocean details, but they are still hard to see and the scale change could be further exaggerated. I did not immediately realize that the vertical scale change was not included in the right panels. I would also appreciate if the Atlantic water was somehow better identified or more visible in this figure – I don't clearly see it in all sections.

Yes, indeed this figure contains a lot of information and some details cannot be identified easily. We made an attempt to improve the presentation. We changed the color of the mixed layer depth to dark green. To better indicate the location of the Atlantic Water, we added a thicker white contour for the 2°C isoline. We also changed the vertical scale of the figure and now zoom in the upper 600 m. The vertical scale is now similar between both right and left panels.

- Figure 4: Some of the subfigure tick-labels are overlapping and hard to read. The legend is small and difficult to read. Also, it may be helpful to replace legend labels with "inshore, shelf break, offshore" as are used in the text. It's difficult to see the details in the upper water column (below ∼100m); you may consider using a different vertical scale (as in figure 3), or providing insets that zoom in on the surface of each panel.

Thanks for all these comments to improve the figure. We now make sure that subfigure tick-labels do not overlap. We changed the legend labels too. These panels are presented to give an overview of the average profiles at depth, and do not aim at focusing in the upper ocean. We therefore do not present split panels with a zoom in on the surface.

- L206-208: Here you say that the core of the Atlantic water current is between 400 m and 600 m, but in L186-187 you associate the Atlantic water with 500 m to 1100 m depths. Throughout the text you use the 800 m isobath as a reference for Altantic water, which is consistent with L186-187 but not with L206-208. Please clarify this and ensure consistency throughout.'

The core of the Atlantic Water current is indeed between 400 and 600 m depth in the water column. L186-187 we stated that the Atlantic Water core is between the 500 and 1100 m depth isobaths (spatial/horizontal location, not vertical location).

- L208: The only mention of current measurements throughout the rest of the paper are the modeled tidal currents, but this sentence is about water column currents. Are these measured with a shipboard ADCP during the cruise? Or is this sentence a reference to known characteristics of the Atlantic water layer from other studies (e.g., the submitted work by Kolås et al., that you reference in L196)?

Yes indeed this sentence is a reference to known characteristics of the Atlantic water layer from other studies. We added the reference to Kolås et al. (2020).

- L250-256: The non-linear dependence of mixed-layer dissipation on wind energy input is a really interesting result, but it would be valuable to explore this concept in more detail and relate it to prior studies (either here, or in some part of the discussion). In particular, there has been some theory that looks at this relation in the wave-boundary layer and may or may not support a linear relationship (see Craig & Banner, 1994, doi: 10.1175/15200485(1994)024<2546:MWETIT>2.0.CO;2 and Thomson, 2016, doi: 10.1175/JPO-D-15-0130.1). It's also been considered in a bulk sense in the mixed layer (i.e., as an efficiency; see Sutherland et al., 2013 doi:10.5194/os-9-597-2013 and references therein; though this is still in the wave-breaking framework). I think there might be some richness in the fact that this analysis suggests a non-linear relationship and worth speculating about why or what that might mean (perhaps stratification or mixed-layer depth play in in some way). Additionally, it may be worth mentioning the wave conditions during the sampling in section 3.1, even if only qualitatively.

Wave measurements were performed during the cruise in September and are shown in Løken, Trygve K., et al. "Wave measurements from ship mounted sensors in the Arctic marginal ice zone."

*arXiv preprint arXiv:1911.07612* (2019). On average the waves were about 1 m (significant wave height) during the September cruise. Unfortunately, we are missing microstructure data in the upper layer where surface wave effects are important. The VMP needs several meters to adjust to free-fall, and the measurements are performed next to the ship, the wake of which contaminates the dissipation measurements. We do not aim to resolve the surface-wave induced processes with our dataset and cannot quantify the role of the wave-boundary layer dynamics on the observed non-linear dependence of mixed-layer dissipation on wind energy input. We added in the manuscript:

In the section 'environmental conditions':
'During the cruise in September, surface gravity waves were estimated using single point ocean surface elevation data obtained from the bow of the ship using a system that combines an altimeter and inertial motion unit (Løken et al., 2019). The significant wave height varied between 0.5 and 1.5 m with mean wave periods between 2 and 6 s."

In the section 'wind forcing':
"During the cruise in September, the surface waves were characterized by 0.5-1.5 m significant wave height (Sect. 3.1, Løken et al., 2019). Because the dissipation measurements are contaminated by the ship's wake in the upper 10 m, we cannot resolve the role of wave-boundary layer dynamics on the vertical structure of dissipation. Since the wave forcing in September was weak, we do not expect a substantial contribution to the observed non-linear dependence of mixed-layer dissipation on wind energy input. However the relatively large values of Dml in July when E10 was large (circles in Figure 6) might be associated with surface waves.'

- Section 4.2: In this section you take all of the data together, but in section 4.1 you contrast some of the details of the mixed-layer between the July and September cruises. Am I correct in interpreting from figure 6 and L252-253 that there's not enough data in September to be able to make meaningful comparisons of Dml between seasons? If it's possible to contrast the effects of wind forcing between the two seasons at all, it would be very interesting.

Indeed, there are so few data points in September in the mixed layer that we are not able to make a meaningful comparison of Dml between seasons.

- Figure 6: Are there any noticeable relationships if you colour the points by mixed-layer depth?

Below is the figure 6 with the points colored by the mixed layer depth. There is no obvious relation that warrants further investigation.

[Figure]

Figure 1. As in Figure 6 but color-coded for the mixed layer depth.

- L316: What are the confidence intervals on the decay scales? Are 18 m and 22 m statistically different from each other?

Both decay scales are statistically different from each other. The 95% confident level is about 2m. We added the confidence interval in the text.

- L320: "We investigate the role of two distinct contributions from tidal currents". Contributions to what? This sentence isn't clear.

We clarify this sentence: 'We investigate the role of two distinct contributions from tidal currents to the turbulent mixing'.

- L326: Why the choice of 250 m for integrating the dissipation? Is this choice informed in any way by the estimated decay scales from earlier? Are results sensitive to other choices?

We chose 250 m as this is the depth where dissipation rate decrease in Figure 8. We tested the sensitivity to different choices (e.g., integrating up to 3 or 5 decay length scales). Although the regression coefficients vary, same trends are observed.

- L330-331: Why exclude wave (tidal) frequency?

We agree that we should have taken into account the tidal wave frequency. We recalculated using the 4 main tidal components (M2, S2, O1, K1), including their corresponding frequencies in the analysis. The contribution from each constituent (using the cross-isobath component of predicted tidal current at the time and location of each station) is summed up to obtain the new data points. The result is shown in Figure 2a. Because we removed these calculations from the paper, we do not describe in detail here. Overall, the new calculations using the frequency dependence are very similar to the original calculations. However, as suggested in the next comment, we decided to remove this panel as the tidal work does not correlate well with D250.

[Figure]

$$D_{250} = (7 \times 10^{-3}) \; X^{(0.16 \pm 0.09)}$$

$$\log_{10} (X = N^2 \, |u_t \cdot \nabla H \, |^2)/\omega \; (m^2 \; s^{-3})$$

Figure 2: Depth-integrated dissipation rate in the bottom 250 m regressed against the instantaneous (using $u_t$ ) values of a tidal-work related parameter, summed over 4 main tidal components (M2, S2, O1, K1). Linear fits on logarithmic parameter space (i.e., power-law fits) are the black lines and the corresponding equations are indicated with the 95% confidence levels. Red dots are the data points from the RS2 station. Process stations are batch-averaged (in sets of 4-5 consecutive profiles) in panel a.

- L327-347: Since the tidal-work parameter in equation 7 doesn't provide a useful correlation, you could choose to simplify the text and figure 9 by removing some of the text in the section, and simply stating that you also tried comparing D250 with the rate of work given by Nash et al., (2006) but found no significant correlation. Then you could remove equation 7 and some of the text surrounding it and remove panels a and b from figure 9. This is a personal choice, but it would better highlight your positive results.

We agree that the tidal-work parameter does not provide a useful correlation. As suggested, we simplified the text and figure 9 and simply stated that D250 and the tidal work do not correlate well.

- L335: It's worth highlighting somewhere that equation 8 is analogous to the equation for E10 (in section 4.2), and so the nonlinear relationship between D250 and Wbotdrag is something that can be related to the nonlinear relationship between Dml and E10.

Yes indeed it is valuable to highlight that equation 8 is analogous to the equation for E10.
Below Eq (6), we inserted:
"Note that this equation is analogous to the drag relation for the wind energy flux E10 (in section 4.2)."

In line 314 we added:
'This nonlinear relationship between D_250 (dissipation in the bottom 250 m) and W_botdrag shows parallels with the nonlinear relationship between D_ml (dissipation in the surface mixed layer) and E_10'

- L350: Does it make sense to compare the bottom drag coefficient to one from the Bering Strait? I wouldn't assume that the bottom morphologies of the two areas would be similar. Can you instead refer to a range of "typical" bottom drag values?

We cite the bottom drag coefficient estimated in the Bering Strait because it is a relevant estimate that was done in the Arctic Ocean from in situ turbulent observations. We now also refer to the typical bottom drag values:

"This value is comparable to but smaller than the typical range of bottom drag values of $(1-3) \times 10^{-3}$ and the bottom drag deduced from in situ observations in Bering Strait… …."

- Figure 9: I don't quite understand why there is only one red dot in panels b and d but many in panels a and c? Is this due to how you perform the u_rms calculation?

The u_rms calculation is performed at a given location, hence the RS2 data points are only one dot in the computation of the u_rms.

- L367-368: If you are showing only a line along the Eurasian Basin, then "Pan-Arctic" in the section title is not appropriate. (Note, the authors have already expressed plans to rename this section).

Yes we agree and we changed the title to 'Estimates of tidally-driven dissipation rate in the Eurasian Basin'.

- L399-400: Maybe highlight to what extent the dissipation in panel b of Figure 10 will account for the nonlinear waves (e.g., if it did account for it, I'd expect to see peaks in D250 in panel b that correspond to the peaks of inverse Fr in panel c). Make it explicit that these are areas that warrant specific further study.

Thanks for this suggestion. We inserted:
'In the region north of Svalbard and in the eastern part of the Laptev Sea, the large depth-integrated dissipation rate observed in Figure 10b can be driven by nonlinear waves implied by the peaks of inverse Fr (Figure 10c). These two areas warrant further studies. In the eastern part of the Kara Sea, however, the depth-integrated dissipation rates are relatively low despite the large inverse Fr values that suggest nonlinear processes could develop there.'

- L387-405: While the discussion of these potential non-linear wave "hotspots" is very interesting, it feels somewhat disconnected from the rest of the study. Most of the times non-linear waves present in the results before this point are references to the high dissipation event at RS2 that was already documented by Fer et al. (2020b). This section would connect more if you make more explicit comparisons between the general results and the non-linear wave results (e.g., you have RS2 points in red in figure 9, which show the associated increase in D250, but those points are presented as more of a sidenote in the text L351-353 when there's potential to make more direct comparisons).

We connected the two sections better by adding a cross-reference to Fig 8 and also given the value for the inverse Fr at RS2. We inserted:
'The increase in dissipation rate driven by these nonlinear waves is also noticeable in Figure 8a and c (the red dots). At this location, the inverse Froude number for the diurnal frequency exceeds 3,

supporting the interpretation that such conditions can favor the development of nonlinear processes.'

- Section 7.1: Overall, this is a nice extension of the ideas in section 6.

Thanks

- Section 7.2: As written, there is no clear link between the ideas in the section and the results you've presented. Do your results agree with or refute any of the studies you cite? Can they be compared at all? This section provides interesting background and motivation, but without linking it explicitly to your results it is not really a discussion section. (Note, the authors have already expressed plans to remove this section).

Indeed, we agree that there is no clear link between the ideas in the section and the results we have presented. For this reason, we decided to remove this section.

- Section 7.3: Similar to section 7.2, this provides good background but isn't otherwise well linked to the rest of the study.

We agree that this section was not well linked to the rest of the study. We reformulated and connected this section better with our results. We revised the last 3 paragraphs:

'Ivanov and Timokhov (2019) estimated that from the Yermak Plateau to the Lomonosov Ridge, 41% of the Atlantic Water heat is lost to atmosphere, 31% to deep ocean and 20% is lost laterally. Heat loss resulting from vertical heat fluxes contributes to the heat loss to atmosphere and to deep ocean, but not to the lateral heat loss. Several processes can lead to lateral heat loss North of Svalbard, including eddy spreading from the slope into the basin (Crews et al., 2018; Våge et al., 2016). Using eddy-resolving regional model results, Crews et al. (2018) found that eddies export 1.0 TW out of the boundary current, delivering heat into the interior Arctic Ocean at an average rate of $\sim$15W m$^{-2}$. West of Svalbard, Kolås and Fer (2018) found that the measured turbulent heat flux in the WSC was too small to account for the cooling rate of the Atlantic Water layer, but reported substantial contribution from energetic convective mixing of an unstable bottom boundary layer on the slope. Convection was driven by Ekman advection of buoyant water across the slope, and complements the turbulent mixing in the cooling process. The estimated lateral buoyancy flux was about 10$^{-8}$ W kg$^{-1}$ (Kolås and Fer, 2018), sufficient to maintain a large fraction of the observed dissipation rates, and corresponds to a heat flux of approximately 40 W m$^{-2}$. We can expect similar processes to extract heat and salt from the Atlantic Water core north of Svalbard. Such processes can explain why turbulent heat fluxes are only responsible for 10% of the Atlantic heat loss north of Svalbard. Furthermore, large heat loss during extreme events should not be ignored. For example, Meyer et al. (2017) found that the average heat flux of about 7 W m$^{-2}$ across the 0$^{\circ}$C isotherm increased during storms, exceeding 30 W m$^{-2}$. During our survey without extreme wind events, the turbulent heat fluxes represent only a small portion of the heat loss of the Atlantic Water.'

- L459-460: The different values of kappa_bot may be a stronger result to highlight in the summary than the different decay scales (or maybe include both?)

We added the values of kappa_bot in the summary: 'The vertical decay scale of the diffusivity is 22 m for those strong tidal currents, compared to 18 m for weaker tidal currents; the bottom diffusivity is larger with strong tidal currents than for weaker ones (1×10$^{-3}$ m$^2$s$^{-1}$ and 7×10$^{-4}$ m$^2$s$^{-1}$ respectively)'

Technical corrections
- L25-26: Awkward grammar/sentence structure in the sentence starting with "The heat reservoir. . .".

We changed the sentence: 'The heat contained in the Atlantic and Pacific origin waters has the potential to melt the entire sea ice if reaching the surface'

- L41-42: Do you have the correct reference for the sentence "Wind-driven momentum input. . ."? Is this meant to reference Rainville and Woodgate (2009) instead of Rainville and Windsor (2008)?

Yes, indeed there was a mistake in the reference. We corrected it.

- L154: "encounter" should be "encountered"

Thanks, corrected

- L317: In the sentence "We use kappa_bot. . .", should that instead be kappa_bg?

Yes indeed it should be kappa_bg, we corrected it.

- L414: Awkward grammar/sentence structure in the sentence starting with "They found. . .".

This section is now deleted.

**Response to referee #1**
This manuscript presents a particularly interesting set of turbulence observations from north of Svalbard in the Arctic that cover the 2018 summer and autumn period. The authors investigate the vertical structure of mixing, heat fluxes and seasonal changes, and identify the processes driving the variability in the turbulence field. Both the wind and tidal supplies of energy are estimated with parameterizations derived and discussed. An attempt to extrapolate to the whole Eurasian Basin is made and interesting areas are identified that could be investigated in further work. The current lack of turbulence measurements in the Arctic is highlighted as the main limitation to pan-Arctic parameterization as well as the difficulty in accounting for lateral processes and fluxes, and for extreme events such as storms. The quality of the English in the text is excellent. The Abstract and Introduction are good, the Data & Methods and Observations sections are excellent. The Upper layer Dynamics section is fine. The section on Mixing in the AW layer is very interesting. The Tidal Mixing section presents a very nice analysis and tools. The Discussion is hard to link to this particular study's findings. The Summary section is excellent. The figures are excellent and have great detail. It was a pleasure to read through this work.

Thanks for these comments

Major comments:
-Introduction: Sort out the introduction part on the various sources and intensity of turbulence in the Arctic (see individual comments further down).

Agreed. We rearranged the introduction as suggested below

-Discussion: Currently the discussion section reads in parts (see individual comments) more like a literature review than a discussion around how your findings fit in current research and their wider impact and implications. You have excellent results and just need to rewrite this section a little. In its current form, the manuscript is already very good and presents a trove of findings for this region on the topic of turbulence. However, the manuscript would benefit from some sorting in parts, better highlight of key findings throughout (done well in the Summary), and better framing of this study's results in the discussion. I recommend that the manuscript is accepted subject to minor revision and look forward to seeing a revised version.

We have rearranged the discussion as suggested below

Individual comments
Abstract:
- Well written overall. The first sentence could do with rewriting to better reflect the beginning of your introduction. Right now you fit too much in that sentence and loose some of the meaning.

We changed the first sentence of the introduction: 'The Arctic Ocean has major implications on global scale as the Arctic Ocean is a main sink for heat and salt. Ocean mixing contribute to this sink by mixing the Atlantic and Pacific-origin waters with surrounding waters.'

1.Introduction:
- L23: You state 'In the near future we may enter a new regime, in which the interior Arctic Ocean is entirely ice free in summer and sea ice is thinner and more mobile in winter'. I would argue that 'may' here is inappropriate and 'will' is more suitable. 'May'creates doubts around the likelihood of this happening. Please rephrase to better reflect current research findings such as the latest estimate from Guarino et al. (Guarino,M., Sime, L.C., Schröeder, D. et al. Sea-ice-free Arctic during the Last Interglacialsupports fast future loss. Nat. Clim. Chang. (2020). https://doi.org/10.1038/s41558-020-0865-2) of 2035 for first ice free summer, or average from CMIP6 models of 2046 with a range of roughly 2030-2065.

We changed 'may' to 'will' and we added the reference Guarino et al., 2020

- L31-37 and L38-46: In both these paragraphs, you describe the various sources and intensity of mixing in the Arctic. These two sections could do with merging and a better ordering of the different sources and intensity discussed.

We merged the two paragraphs and we ordered better the sources and intensities. We also removed part of the description of the sources and intensities as we found that it did not serve the rest of the manuscript.

- L65: Consider adding the following reference somewhere here: 'The lack of sea ice is mainly due to heat from the Atlantic layer reaching the surface'. Duarte, P., Sundfjord A., Meyer, A., Hudson, S. R., Spreen, G., & Smedsrud, L. H. (2020). Warm Atlantic water explains observed sea ice melt rates north of Svalbard. Journal of Geophysical Research: Oceans, 125, e2019JC015662. https://doi.org/10.1029/2019JC0156622.

We added the reference

2.Data and Methods:

- L105: Unclear what 'In total, we collected 31 profiles.' Do you mean ship CTD profiles? Or VMP profiles or ? This doesn't match other number of VMP profiles stated earlier in the manuscript.

Thanks for spotting this mistake. We deleted this sentence

- L126: Pls define 'g' in equation (4) if not previously defined.

We added the definition of g: 'where alpha and beta are respectively the thermal expansion and salinity contraction coefficients, and g is the gravitional constant.'

L129-130: You state here that 'We used the profiles collected from the ship's CTD system (Sea-Bird Scientific,SBE 911plus on both cruises) to check and correct the temperature and salinity from the VMP'. But earlier on L107 you state 'A good agreement was observed and no correction was made.'. Please rewrite to make both statement consistent.

Thanks for pointing it out. We deleted 'correct' in the first sentence.

3.Overview of observations:
- L172-173: Unsure you need this statement here considering you have explained it clearly it in the figure caption.

Agreed. We deleted this sentence

- Figure 3: Add what the red line is MLD in the caption.

We added in the caption that the (now) green line is the mixed layer depth.

4. Upper layer dynamics:
- L252: Add the definition of Dml in the text. Currently it only appears in Fig.6 caption. Can you make it clearer in the text how you obtained your estimate of the relationship between Dml and E10: it's a linear fit of Dml from the VMP data and E10 from the shipwind speed measurements.

We added the definition of Dml and clarify that we apply a linear fit

5. Mixing in the Atlantic Water layer:
- L264: Should 'in present conditions of a warming Arctic' not be 'in the new conditions of a warming Arctic'?

Changed as suggested.

- Fig.7 is great

Thanks!

- L274-275: This statement is confusing 'vertical turbulent heat fluxes are negative(less than 5Wm-2)' You might want to rephrase to 'vertical turbulent heat fluxes are negative (0 to -5Wm-2)'

Changed as suggested.

- L282: Which section are you speaking about when you say '...the heat loss due to vertical turbulent heat fluxes is about... across the section'?

We are talking about the cross-isobath section. We agree that 'across the section' is more confusing than helpful and we deleted it.

- L282-285: Why is your estimate of heat loss due to vertical turbulent heat fluxes(1.2x10^5 W/m) so much lower than Kolas estimates from the same cruise (9.1x10^7W/m and 9.6x10^6 W/m)?

Here we estimate the heat loss only due to vertical turbulent heat fluxes. Kolås et al. (2020) estimate the along-path change of heat content, that takes into account not only the vertical turbulent heat fluxes but also the other fluxes that can impact the heat content.

6. Tidal mixing:
- Fig. 8 caption: 'Average profiles of a) dissipation rate, b) turbulent heat flux and c) diapycnal diffusivity k for small' Also add Espi and F_H after the variable's names.

Done

- L326-366: Nice analysis of the vertically integrated dissipation rate in bottom 250m.

Thanks

7. Discussion:
- Fig.10 caption: I suggest removing the first word 'Typical'. Also, what is the back-ground shading on the small map, topography? This map is useful and should be listed in the caption.

'Typical' reinforce the idea that we use u_rms. The background shading on the small map is topography, we added this information in the caption.

- L358: Subsection title 'Pan-Arctic estimates of tidally-driven dissipation rates' is not representative of results presented which are 'instead of presenting Arctic-wide maps we concentrate on the Eurasian Basin from north of Svalbard into the East Siberian Sea'. Please change section title to represent better the content. Also edit L355 in the previous section announcing the 'pan-Arctic estimate'.

We changed the title of the subsection to 'Estimates of tidally-driven dissipation rate in the Eurasian Basin' and we edited l.355: 'An Eurasian-basin coarse estimate will be given ...'.

- L398-405: Great findings.

Thanks

- L410: Rephrase sentence 'In the future, sea ice meltwater is expected to increase and turbulent mixing near the surface to decrease' to better justify/explain the expected decrease in mixing (due to increase stratification).
- L 423: 'and an earlier onset of stratification which might be indirectly linked to bloom development'...due to.... Please add details.

- Section 7.2: I m unsure about the contribution your results make in this theme of 'impact of meltwater on the near surface mixing'. Consider better linking to your observations or moving this section as context in your introduction in a condensed form.

We agree that this discussion is not really relevant to our analysis. We deleted section 7.2.

- L433: I m unsure about how this statement 'Vertical turbulent heat fluxes are not the main source of cooling of the Atlantic Water layer in the Arctic. Ivanov and Timokhov (2019) reviewed that from the Yermak Plateau to the Lomonosov ridge, 41% of the Atlantic Water heat is lost to the atmosphere, 31% to the deep ocean and 20% is lost laterally.' fits with the previous 'heat loss due to turbulent vertical mixing represents less than 10% of the total heat loss of the Atlantic Water' . Would the 10% not be part of the 31% deep ocean and 20% laterally? You seem to imply they are different when you state 'Vertical turbulent heat fluxes are not the main source of cooling of the Atlantic Water layer in the Arctic'. Please tidy up these two paragraphs so the reader can follow your thoughts. Again, further down you discuss eddies and their roles. But is the heat export from eddies not included in the 20% lost laterally from Ivanov and Timokhov (2019)?

Yes, you are right. We are mixing different informations. We found that turbulent vertical mixing represents less than 10% of the total heat loss of the Atlantic Water layer, but indeed we do not specify where the heat is lost, so these 10% are not to be compared with the percentages from Ivanov and Timokhov (2019). We changed the sentence: '
Ivanov and Timokhov (2019) estimated that from the Yermak Plateau to the Lomonosov Ridge, 41% of the Atlantic Water heat is lost to atmosphere, 31% to deep ocean and 20% is lost laterally. Heat loss resulting from vertical heat fluxes contributes to the heat loss to atmosphere and to deep ocean, but not to the lateral heat loss. '

- L444 and 445: The numbers you quote there (10ˆ-8 and 40W/mˆ2), are they from Kolas and Fer or from this study? Again, how does this section of the discussion(7.3 AW heat loss) exactly links with your findings. Currently this reads a lot like an (excellent) literature review, rather than you putting your new findings in context...

These numbers were from Kolås and Fer. We agree that this section looks more like a literature review, and we tried to better put our new findings in context. We mainly changed the last 2 paragraphs:

'West of Svalbard, Kolås and Fer (2018) found that the measured turbulent heat flux in the WSC was too small to account for the cooling rate of the Atlantic Water layer, but reported substantial contribution from energetic convective mixing of an unstable bottom boundary layer on the slope. Convection was driven by Ekman advection of buoyant water across the slope, and complements the turbulent mixing in the cooling process. The estimated lateral buoyancy flux was about $10^{-8}$ W $kg^{-1}$ (Kolås and Fer, 2018), sufficient to maintain a large fraction of the observed dissipation rates, and corresponds to a heat flux of approximately 40 W $m^{-2}$. We can expect similar processes to extract heat and salt from the Atlantic Water core north of Svalbard. Such processes can explain why turbulent heat fluxes are only responsible for 10% of the Atlantic heat loss north of Svalbard. Furthermore, large heat loss during extreme events should not be ignored. For example, Meyer et al. (2017) found that the average heat flux of about 7 W $m^{-2}$ across the $0°C$ isotherm increased during storms, exceeding 30 W $m^{-2}$. During our survey without extreme wind events, the turbulent heat fluxes represent only a small portion of the heat loss of the Atlantic Water.'

8. Summary:

- L459-460: Consider adding 'The vertical decay scale of the diffusivity is 22m *for those strong tidal currents*, compared to 18m for weaker tidal currents.'

Thanks, done

- L470: Consider adding details 'More in situ observations from different sites *in the Eurasian Basin and elsewhere in the Arctic* are needed to confirm our results.'

Thanks, done

- L475: Can you add 'of the *expected/estimated* total heat loss of the Atlantic Water layer'.

We added 'estimated'

- L475-476: Can you explain better the relation between the first part of the sentence and the later part? I understand you mean to say that increased vertical mixing during storms might partially close the budget but don't make up the whole 'missing' heat loss which might be mostly lateral fluxes. So that both lateral fluxes and extreme conditions such as storms, frontal systems etc should be investigated. But this will not super clear in the current form of the sentence.

We reformulate the last sentence: 'Increased vertical mixing during storms would add to this figure. However, integrated studies addressing lateral mixing processes, frontal systems as well as extreme conditions such as storms are needed to close the heat budget in this region.'